# Hepatocyte-derived IL-10 plays a crucial role in attenuating pathogenicity during the chronic phase of *T. congolense* infection

**Benoit Stijlemans**[1,2]*, **Hannelie Korf**[3], **Patrick De Baetselier**[1,2], **Lea Brys**[1,2], **Jo A. Van Ginderachter**[1,2☉], **Stefan Magez**[1,4☉], **Carl De Trez**[1☉]

**1** Lab of Cellular and Molecular Immunology, Vrije Universiteit Brussel (VUB), Brussels, Belgium, **2** Myeloid Cell Immunology Lab, VIB Center for Inflammation Research, Brussels, Belgium, **3** Laboratory of Hepatology, Department of Chronic Diseases, Metabolism and Ageing (CHROMETA), KU Leuven, Leuven, Belgium, **4** Laboratory for Biomedical Research, Ghent University Global Campus, Incheon, South Korea

☉ These authors contributed equally to this work.
* benoit.stijlemans@vub.be

**Data Availability Statement:** All relevant data are within the manuscript and its Supporting Information files.

## Abstract

Bovine African Trypanosomosis is an infectious parasitic disease affecting livestock productivity and thereby impairing the economic development of Sub-Saharan Africa. The most important trypanosome species implicated is *T. congolense*, causing anemia as most important pathological feature. Using murine models, it was shown that due to the parasite's efficient immune evasion mechanisms, including (i) antigenic variation of the variable surface glycoprotein (VSG) coat, (ii) induction of polyclonal B cell activation, (iii) loss of B cell memory and (iv) T cell mediated immunosuppression, disease prevention through vaccination has so far been impossible. In trypanotolerant models a strong, early pro-inflammatory immune response involving IFN-γ, TNF and NO, combined with a strong humoral anti-VSG response, ensures early parasitemia control. This potent protective inflammatory response is counterbalanced by the production of the anti-inflammatory cytokine IL-10, which in turn prevents early death of the host from uncontrolled hyper-inflammation-mediated immunopathologies. Though at this stage different hematopoietic cells, such as NK cells, T cells and B cells as well as myeloid cells (*i.e.* alternatively activated myeloid cells (M2) or Ly6c⁻ monocytes), were found to produce IL-10, the contribution of non-hematopoietic cells as potential IL-10 source during experimental *T. congolense* infection has not been addressed. Here, we report for the first time that during the chronic stage of *T. congolense* infection non-hematopoietic cells constitute an important source of IL-10. Our data shows that hepatocyte-derived IL-10 is mandatory for host survival and is crucial for the control of trypanosomosis-induced inflammation and associated immunopathologies such as anemia, hepatosplenomegaly and excessive tissue injury.

**Funding:** This work was performed in frame of an Interuniversity Attraction Pole Program (PAIIAP N. P7/41, http://www.belspo.be/belspo/iap/index_en. stm) and was supported by the Strategic Research Program (SRP3 and SRP47, VUB) and a grant from the FWO (FWO G015016N). BS was supported by the Strategic Research Program (SRP3 and SRP47, VUB). JAVG is a member of the EU-COST action Mye-EUNITER. The funders had no role in study design, data collection and analysis, decision to publish, or preparation of the manuscript.

**Competing interests:** The authors have declared that no competing interests exist.

## Author summary

Bovine African Trypanosomosis is a parasitic disease of veterinary importance that adversely affects the public health and economic development of sub-Saharan Africa. The most important trypanosome species implicated is *T. congolense*, causing anemia as most important pathological feature and major cause of death. Using murine models, it was shown that the disease is characterized by a well-timed and balanced production of pro-inflammatory cytokine promoting factors followed by an anti-inflammatory response, involving IL-10. The latter is required to attenuate infection-associated pathogenicity and to prevent early host death from uncontrolled hyper-inflammation mediated immunopathologies. However, the cellular source of IL-10 *in vivo* and the window within which these cells exert their function during the course of African trypanosomiasis remain poorly understood, which hampers the design of effective therapeutic strategies. Using a *T. congolense* infection mouse model, relevant for bovine trypanosomosis, we demonstrate that during the chronic stage of infection hepatocyte-derived IL-10, but not myeloid cell-derived IL-10, regulates the main infection-associated immunopathologies and ultimately mediates host survival. Hence, strategies that tilt the balance of hepatocyte cytokine production in favor of IL-10 could majorly impact the wellbeing and survival of *T. congolense*-infected animals. Given the unmet medical need for this parasite infection, our findings offer promise for improved treatment protocols in the field.

## Introduction

African trypanosomes are extracellular protozoan parasites transmitted by the bite of infected tsetse flies (genus *Glossina*), causing sleeping sickness in humans and Nagana disease in cattle in sub-Saharan Africa. About 60 million people are at risk and Nagana causes three million cattle deaths every year due to fever, weight loss and anemia. By affecting the agricultural production and animal husbandry, Animal African trypanosomosis (AAT) has a high socio-economic impact in vast areas of the tropics and subtropics where the transmission occurs [1–3]. As such, it is considered to be the livestock disease with the highest impact on agricultural production and animal husbandry in Africa, whereby the annual economic loss in livestock production is estimated at 4 billion US$ [1]. *Trypanosoma congolense* is considered to be the most pathogenic trypanosome species in cattle [4]. Although anemia is the most prominent pathological feature of AAT [2,5], sporadic episodes of fever as well as leukopenia, weight loss and hepatosplenomegaly, in conjunction with appetite loss, lethargy and emaciation, can contribute to death through eventual congestive heart failure. Importantly, trypanotolerance in cattle has been described as the capacity of an animal to better control parasitaemia and to have a better capacity to limit anemia development and weight loss, whereby anaemia control is more important for survival and productivity than parasite control [5,6]. Murine models are considered valuable tools to study the interactions between parasites and hosts that contribute to immuno-pathogenicity and allow discriminating between trypanosusceptible versus trypanotolerant animals. Experimental *T. congolense* infections in mice have shown that C57BL/6 mice are able to control the first peak of parasitemia and develop a chronic infection lasting for 4 months, and are therefore considered as relatively trypanotolerant animals [7], while trypanosusceptible BALB/c mice die within 10 days post infection [7]. The infection is characterized by two stages; during the early stage there is a strong inflammatory immune response mediated by T cells and involving classically IFN-γ-activated myeloid cells (so-called M1) required for the efficient control of the first most prominent parasitemia peak through their production of trypanotoxic molecules, such as

nitric oxide (NO) and Tumor Necrosis Factor (TNF), and the phagocytosis of antibody-opsonized parasites that occurs mainly in the liver [8–12]. This is followed by the production of the anti-inflammatory cytokine IL-10, which is essential to dampen the inflammatory immune response after parasitemia has been cleared and to prevent tissue damage as well as death of the host due to a hyper-inflammation syndrome [13–16]. Hence, IL-10 allows the development of a so-called "trypanotolerant" phenotype [11,17]. Accordingly, IL-10-deficient C57BL/6 mice infected with *T. congolense* develop a severe inflammatory response-like syndrome due to excessive production of inflammatory cytokines such as TNF and IFN-γ, resulting in a drastic reduction in survival time [14,17–19]. Conversely, absence of the upstream regulator of the inflammatory cascade, *i.e.* macrophage migration inhibitory factor (MIF), has been shown to correlate with an attenuated inflammatory immune response and a concordantly reduced immunopathology during the course of infection, which is associated with an extended survival time [20]. This phenotype coincides with increased systemic levels of IL-10 during the later stages of infection [20]. Hence, a well-timed and balanced order of pro-inflammatory cytokine promoting factors followed by an anti-inflammatory response is required to attenuate infection-associated pathogenicity. However, the *in vivo* cellular source of IL-10 and the window within which these cells exert their function during the course of African trypanosomiasis, as well as the associated molecular mechanism(s) implicated in its production, remain poorly understood. Hence, knowledge about the inflammation resolution process is necessary to understand the host-parasite interplay and might pave the way to improve or develop more efficient therapies that reduce the devastating effect of chronic protozoan infections [21].

Here, we aimed at refining the cellular contribution of IL-10 during experimental African trypanosomiasis using the *T. congolense* model in C57BL/6 mice. Thus far, both naturally occurring CD4+ Foxp3+ Tregs as well as myeloid cells (Ly6C- patrolling monocytes and alternatively activated macrophages (M2)) were found to be sources of IL-10, that limit to some extent the development of immuno-pathogenicity (including liver injury) during infection [14,15,19]. However, it remains unclear if non-hematopoietic cells, such as hepatocytes, could also be a potential source of IL-10 during the course of the infection. Using IL-10 reporter mice, we confirmed that hepatocytes can produce IL-10 upon exposure to *T. congolense* parasites or parasite-derived products *in vitro* as well as during *T. congolense* infection. Therefore, we investigated the *in vivo* anti-inflammatory potential of hepatocyte-derived IL-10 during *T. congolense* infection, using hepatocyte-specific IL-10-deficient mice. We demonstrate that hepatocyte-derived IL-10 regulates the main infection-associated immunopathologies, such as anemia, weight loss, chronic systemic inflammation, hepatosplenomegaly and liver damage during chronic infection with *T. congolense*. Hence, although other cells can produce IL-10 during experimental *T. congolense* infection, hepatocyte-derived IL-10 is crucial to control inflammation-induced immunopathogenicity during the chronic stage of infection that ultimately mediates host survival.

## Results

### Hepatocyte-specific IL10-deficiency correlates with reduced survival, increased tissue pathogenicity and increased systemic inflammation during *T. congolense* infection

Considering the essential role of IL-10 in limiting the pathogenicity and, thus, the susceptibility to infection, we evaluated if besides leukocytes (*i.e.* regulatory T cells and myeloid cells) also hepatocytes could be a potential source of IL-10 during African trypanosome infection. Therefore, we investigated in more detail the role of hepatocyte-derived IL-10 in the outcome of experimental *T. congolense* infection, a model in which the systemic levels of IL-10 progressively increase during the course of infection in order to sustain host survival [19]. Using IL-10-eGFP

reporter (Vert-X) mice, we confirmed that besides leukocytes (S1 Fig), also hepatocytes from chronically *T. congolense* infected animals can produce IL-10 (Fig 1A–1C). In addition, IL-10 was detected in the supernatant of hepatocyte cultures from chronically *T. congolense* infected wild type (WT) but not hepatocyte-specific IL-10-deficient (TgAlbCre-IL10$^{-/-}$) mice (Fig 1D). Interestingly, induction of IL-10 was observed in *in vitro* cultures of hepatocytes from non-infected WT mice stimulated with either trypanosomes or trypanosome lysate, as well as soluble VSG (sVSG) (Fig 1E), indicating a direct parasite effect. Finally, isolated hepatocytes from chronically *T. congolense* infected WT mice exhibited, besides an increase in *IL-10* gene expression, also an increased *IL-10r* expression compared to hepatocytes from uninfected animals (Fig 1F–1G). In contrast, no increase in *IL-10r* expression was observed in isolated hepatocytes from chronically *T. congolense* infected TgAlbCre-IL10$^{-/-}$ mice compared to hepatocytes from uninfected animals (Fig 1G).

Since myeloid cells could also be a source of IL-10 in infected animals [15], we included in subsequent experiments myeloid-specific IL-10-deficient (LysM-IL-10$^{-/-}$) mice and compared their infection parameters to TgAlbCre-IL10$^{-/-}$ and WT controls. Although early peak parasitemia was similar in all mouse strains, TgAlbCre-IL10$^{-/-}$ mice exhibited significantly higher parasitemia levels during the later stage of infection as well as an increased weight gain (due to increased spleen and liver weight), more severe anemia and a significantly reduced median survival time (Fig 2A–2D, median survival time WT: 102 ± 26; LysM-IL-10$^{-/-}$: 100 ± 14, TgAlbCre-IL10$^{-/-}$: 50 ± 10 days). To confirm that the observed differences were not due to intrinsic gross abnormalities in the TgAlbCre-IL10$^{-/-}$ mice, these mice were infected with another strain of trypanosomes, namely *T. brucei*, which induces a more acute infection with low systemic levels of IL-10 [22,23]. As shown in S2 Fig, compared to WT mice, TgAlbCre-IL10$^{-/-}$ mice exhibited a similar parasitemia, anemia and weight loss profile and had a similar survival profile in this model.

In order to delineate the window within which the contribution of hepatocyte IL-10 plays a role, a kinetic study was performed in these 3 different mouse lines, whereby the blood was investigated with respect to cytokines and white blood cell (WBC) composition (see gating strategy S3A Fig) that can contribute to pathogenicity. As shown in Fig 3, TgAlbCre-IL10$^{-/-}$ mice exhibited similar early cytokine levels as WT and LysM-IL-10$^{-/-}$ mice. Yet, between 38–48 days post infection, all tested pro-inflammatory cytokines (IFN-γ, TNF, IL-6 and MIF) increased or remained high in TgAlbCre-IL10$^{-/-}$ mice, but not in the other strains.

At the level of the WBCs, we observed a progressive increase in the percentage of CD45$^+$ cells during the course of infection in all groups of mice (Fig 4A). A gradual decrease in the absolute number of B cells (per mL blood) was observed in all groups of mice, yet TgAlbCre-IL10$^{-/-}$ mice exhibited significantly lower numbers of B cells between day 38–48 post infection compared to WT and LysM-IL-10$^{-/-}$ mice (Fig 4B). At the level of the T cells, no differences were recorded for any of the groups. Regarding the Ly6C$^{high}$ monocytes, which play a key role in pathogenicity development, we observed a similar increase in all groups as the infection progresses, whereby these cells already exhibited a significant upregulation of MHC-II at day 7 p.i. Interestingly, at the later stages of infection their MHC-II expression was significantly higher in TgAlbCre-IL10$^{-/-}$ mice compared to WT and LysM-IL-10$^{-/-}$ mice (Fig 4C), indicative of a higher monocyte activation level. Likewise, the number of polymorphonuclear cells (PMN), which were also documented to contribute to trypanosome-infection associated pathogenicity [22], were found to progressively increase in all groups as the infection progresses, yet reaching significantly higher levels in TgAlbCre-IL10$^{-/-}$ mice at the later infection stages than in WT and LysM-IL-10$^{-/-}$ mice (Fig 4B). Finally, the number of Ly6C$^{low}$ patrolling monocytes, which were documented to attenuate tissue injury [24], significantly dropped in TgAlbCre-IL10$^{-/-}$ mice at later infection stages (Fig 4B). Overall, TgAlbCre-IL10$^{-/-}$ mice clearly display an increased

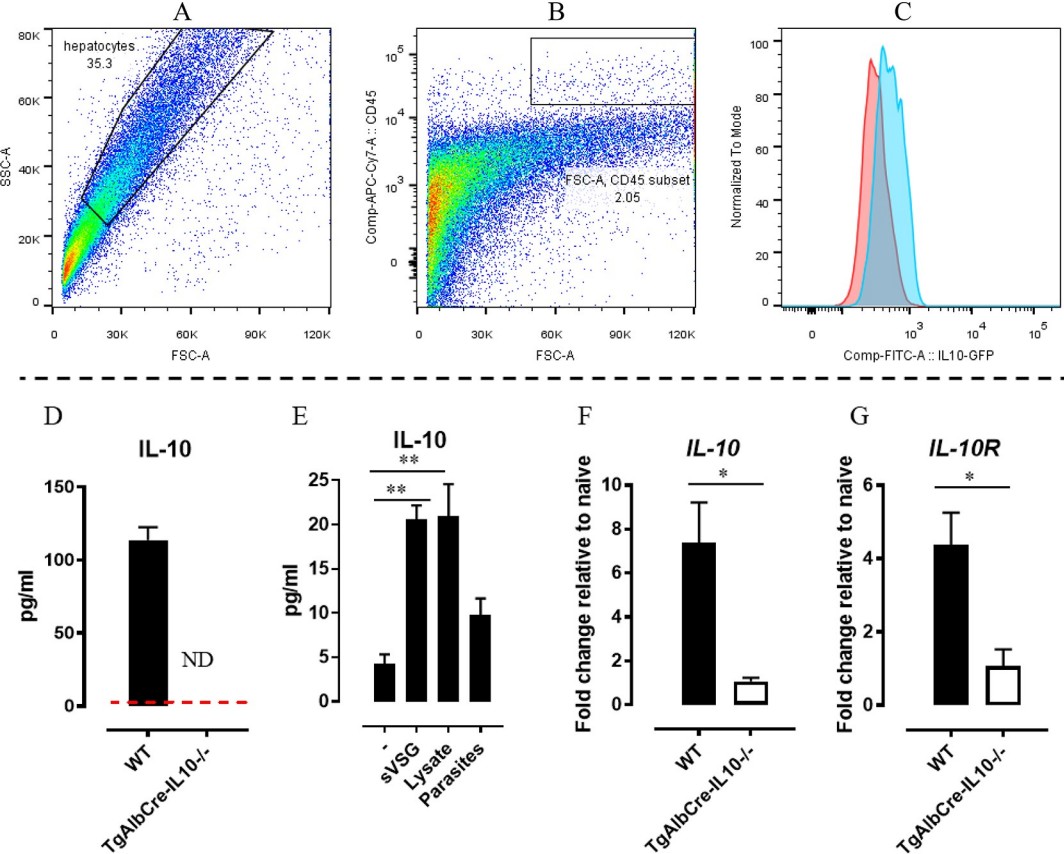

**Fig 1. Hepatocytes from chronically *T. congolense* infected mice are proficient IL-10 producing cells and hepatocytes from non-infected animals can be triggered by trypanosomes to produce IL-10.** Representative FACS profile of purified hepatocytes at day 45 post *T. congolense* infection showing an FSC-A versus SSC-A plot (A) and a CD45 versus FSC-A (B) plot (revealing a purity of more than 95%), whereby hepatocytes were selected based on their CD45⁻ profile. Of note, the selection of the hepatocytes (CD45⁻ cells) is based on the gating strategy used in S1A Fig, whereby debris/death cells can be excluded. (C) Histogram plot showing the intensity of the IL-10-eGFP signal in hepatocytes from IL-10-eGFP reporter (blue) mice and, as negative control, in hepatocytes from TgAlbCre-IL10⁻/⁻ (red) mice. At day 45 post *T. congolense* infection, isolated hepatocytes from WT (black symbol) and TgAlbCre-IL10⁻/⁻ (white symbol) mice were cultured for 36 hours and subsequently tested in ELISA for IL-10 protein levels (D). IL-10 secretion by purified hepatocytes from naïve WT (black bar) following stimulation with $10^7$ parasites or 50 µg lysate or 5 µg sVSG for 36 hours at 37˚C in a 5% $CO_2$ incubator (E). For each condition $2.10^6$ liver cells per ml were used. As controls cells were left untreated. Results are representative of 2 independent experiments and shown as mean of 3 individual mice ± SEM. At day 45 post *T. congolense* infection, isolated hepatocytes from WT (black symbol) and TgAlbCre-IL10⁻/⁻ (white symbol) mice, were tested in RT-PCR for *IL-10* and *IL-10R* gene expression (F and G, respectively). Of note, RT-PCR results are presented as fold change whereby the expression levels were normalized using *S12* and expressed relatively to the expression levels in the corresponding non-infected animals. Non-infected animals as well as TgAlbCre-IL10⁻/⁻ mice did not show any detectable IL-10 protein levels (Dashed line). Data are represented as mean of at least 3–5 mice per group ± SEM and are representative of 2 independent experiments. (*: $p \leq 0.05$, **: $p \leq 0.01$, ND: Not detected).

inflammatory immune response at later stages of *T. congolense* infection, coinciding with reduced B cell and patrolling monocyte numbers that are required for parasite control and attenuation of tissue injury, respectively.

Since the largest differences in pathogenicity (*i.e.* anemia and weight change) and blood parameters were established between 38–48 days post infection, a more refined analysis was performed around day 45 post *T. congolense* infection. This time point was selected for all further experiments. Considering the crucial role of IL-10 in attenuating the pathogenic effects of pro-inflammatory cytokines, the reduced survival of TgAlbCre-IL10⁻/⁻ mice could be due to a higher tissue pathogenicity in association with higher levels of these cytokines [14,19]. In agreement, as compared to WT and LysM-IL-10⁻/⁻ mice, TgAlbCre-IL10⁻/⁻ mice exhibited

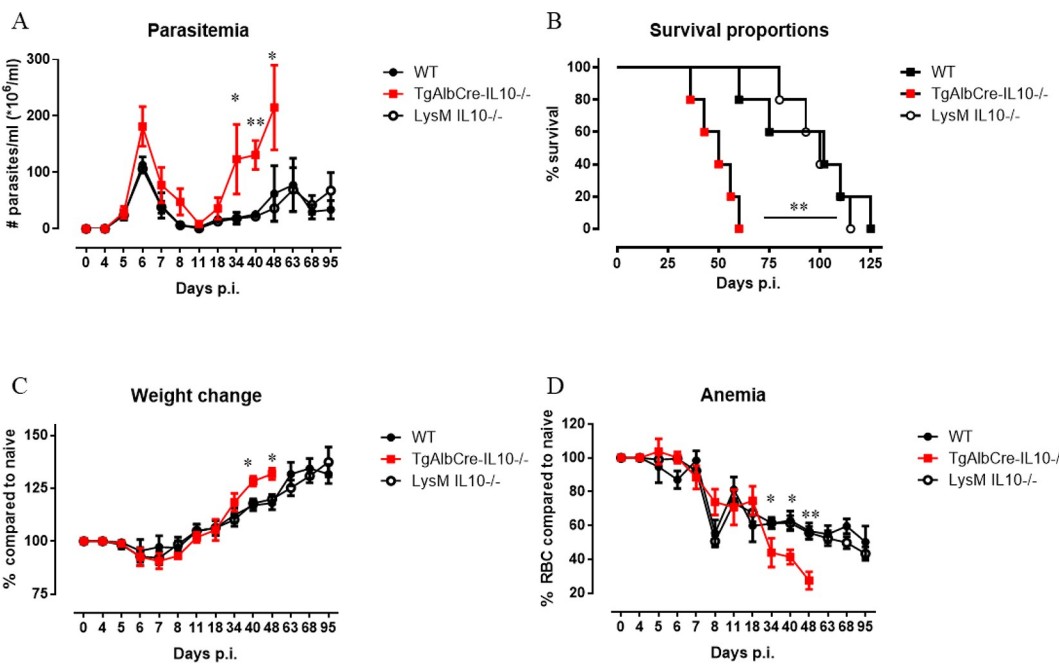

**Fig 2. Hepatocyte-specific IL10-deficiency correlates with reduced survival and increased anemia during *T. congolense* infection.** A) Parasitemia, (B) Survival, (C) weight change and (D) anemia of *T. congolense* infected wild type (WT, black symbol), LysM-IL-10$^{-/-}$ (white symbol) and TgAlbCre-IL10$^{-/-}$ (red symbol) mice. Data are represented as mean (A, C-D) or median (B) of 3–5 mice per group ± SEM and are representative of 2–3 independent experiments. (*: p≤0.05, **: p≤0.01).

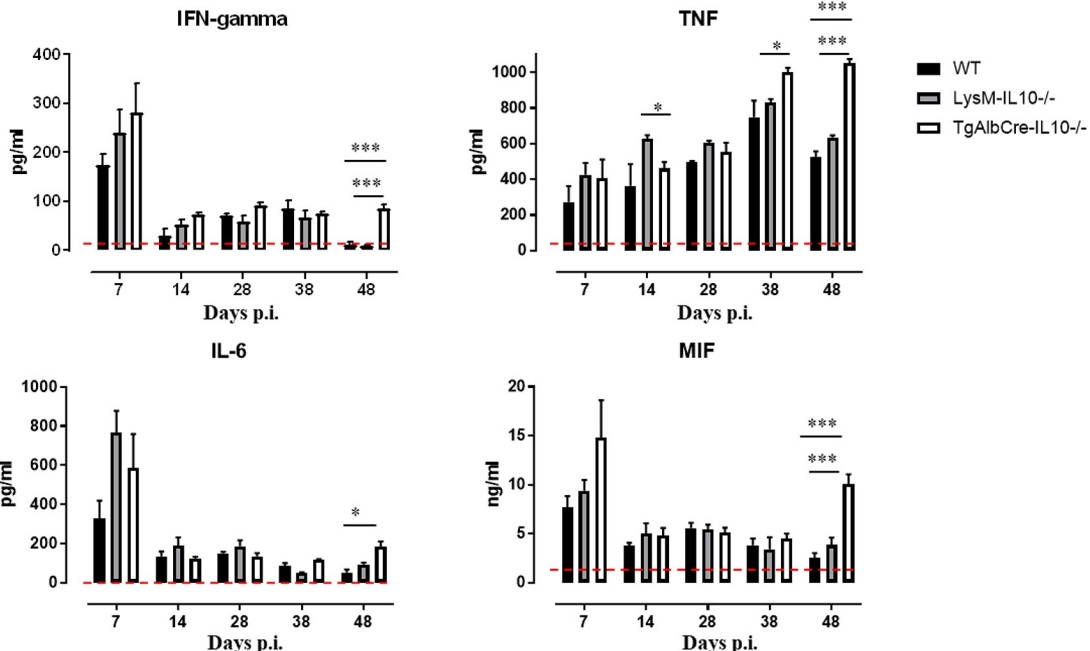

**Fig 3. Hepatocyte-specific IL10-deficiency correlates with increased pro-inflammatory serum cytokine levels during the chronic stage of *T. congolense* infection.** Serum cytokine kinetics of *T. congolense* infected wild type (WT, black symbol), LysM-IL-10$^{-/-}$ (grey symbol) and TgAlbCre-IL10$^{-/-}$ (white symbol) mice. Dashed line represents cytokine levels in non-infected animals. Data are represented as mean of 5 mice per group ± SEM and are representative of 2 independent experiments. (*: p≤0.05, **: p≤0.01, ***: p≤0.005).

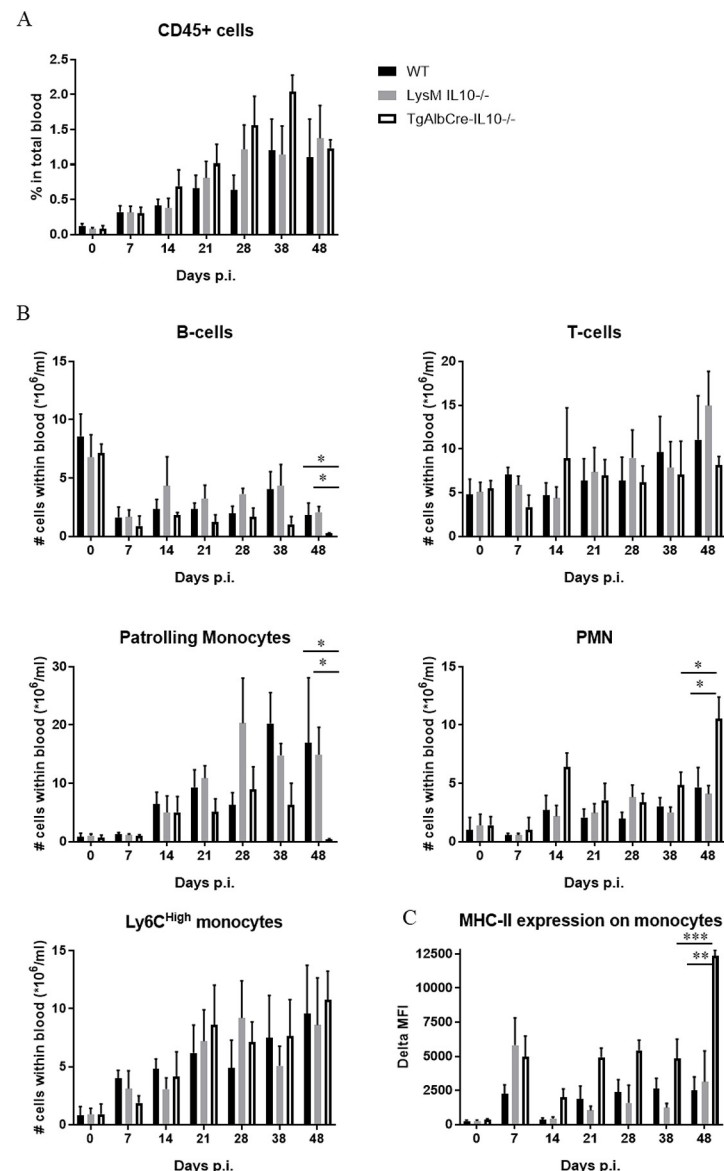

**Fig 4. Hepatocyte-specific IL10-deficiency correlates with decreased B cell and patrolling monocyte numbers, increased MHC-II expression on inflammatory monocytes and increased PMN numbers in the blood during the later stage of *T. congolense* infection.** Kinetic of (A) the percentage of CD45$^+$ cells, (B) the number of white blood cells (WBC, including B cells, T cells, patrolling monocytes, polymorphonuclear cells (PMN) and inflammatory monocytes) and (C) the MHC-II expression levels on inflammatory monocytes expressed in delta median fluorescence intensity (delta MFI) during the course of *T. congolense* infection in wild type (WT, black symbol), LysM-IL-10$^{-/-}$ (grey symbol) and TgAlbCre-IL10$^{-/-}$ (white symbol) mice. The gating strategy is described in S3A Fig. Data are represented ± SEM (5 mice per group) and are representative of 2 independent experiments. (*: p≤0.05, **: p≤0.01, ***: p≤0.005).

increased serum AST (aspartate aminotransferase, reflecting systemic tissue injury) and ALT (alanine aminotransferase, reflecting liver injury) levels as well as creatinine (reflecting kidney injury) levels (Fig 5A–5C). In TgAlbCre-IL10$^{-/-}$ mice, this observation coincides with increased serum levels of pro-inflammatory cytokines (IFN-γ, TNF, IL-12p70, IL-6 and MIF) which were documented to contribute to *T. congolense*-induced tissue destruction (Fig 5D)

[20,25–27]. In line with previous observations, LysM-IL-10[-/-] mice also showed higher systemic levels of TNF compared to WT mice [15].

Collectively, we observed that parasites can induce IL-10 production by hepatocytes and that hepatocyte-derived IL-10 is crucial to attenuate systemic inflammation and *T. congolense* infection-associated pathogenicity during the later stages of infection resulting in a prolonged survival.

## Hepatocyte-specific IL10-deficiency correlates with increased hepatosplenomegaly and an altered cellular tissue composition during *T. congolense* infection

During chronic infections with trypanosomatids, including African trypanosomes, hepatosplenomegaly develops as an additional complication associated with the disease [20,22,23,28]. This pathological feature results in the overall increase in *T. congolense* infection-associated mouse body weight (Fig 2C) [20,29]. Notably, though *T. congolense* infections induce hepatosplenomegaly in all mouse strains used in this study, this feature was most pronounced in TgAlbCre-IL10[-/-] mice (Fig 6A and 6B) correlating with a higher mouse weight. However, when subtracting the level of hepatosplenomegaly (*i.e.* the weight of liver and spleen) from the actual mouse weight it appears that *T. congolense* infected animals, compared to non-infected animals, loose weight (which is also observed in cattle [30]) (S4 Fig). In this context, TgAlbCre-IL10[-/-] mice showed an earlier and significantly higher weight loss compared to the other groups.

Remarkably, the number of liver-associated cells, comprising both hepatocytes (CD45[-] cells) and liver-associated leukocytes (CD45[+] cells), significantly increased in all infected animals compared to non-infected animals, with no further increase in TgAlbCre-IL10[-/-] mice (Fig 6C) despite the significantly bigger liver size in these animals (Fig 6A and 6B). However, livers from TgAlbCre-IL10[-/-] mice contained significantly lower numbers of hepatocytes but tended to have higher numbers of leukocytes compared to WT and LysM-IL-10[-/-] mice. Also the spleens of all infected mice contain more WBC compared to non-infected mice, but, similar to the liver, TgAlbCre-IL10[-/-] spleens do not contain more WBC than WT or LysM-IL-10[-/-] spleens (Fig 6D). Conversely, spleens from infected TgAlbCre-IL10[-/-] mice encompass significantly higher numbers of red blood cells (RBCs, Fig 6E, see gating strategy [20]) compared to infected WT and LysM-IL-10[-/-] mice. Hence, the enhanced splenomegaly observed in TgAlbCre-IL10[-/-] mice might be due to a higher level of extramedullary erythropoiesis, a phenomenon which was reported before to cause splenomegaly in *T. congolense* infected animals [20,29].

Collectively, the absence of hepatocyte-derived IL-10 results in a more pronounced hepatosplenomegaly, which correlated with an altered tissue cellular composition.

## Hepatocyte-specific IL10-deficiency correlates with enhanced erythrophagocytosis, increased hemodilution and thrombocytopenia during *T. congolense* infection

We have shown before that *T. congolense* infection causes hemodilution, due to the low yet chronic nature of the inflammation, which coincides with the occurrence of hepatosplenomegaly [20]. Interestingly, infected TgAlbCre-IL10[-/-] mice exhibited more severe hemodilution compared to WT and LysM-IL-10[-/-] mice as illustrated by an enhanced blood (Fig 7A) and plasma (Fig 7B) volume. This increased hemodilution resulted in thrombocytopenia, encompassing a decrease in platelet counts per blood volume and in the absolute platelet number in the blood of the respective mice, which is again most pronounced in TgAlbCre-IL10[-/-] mice

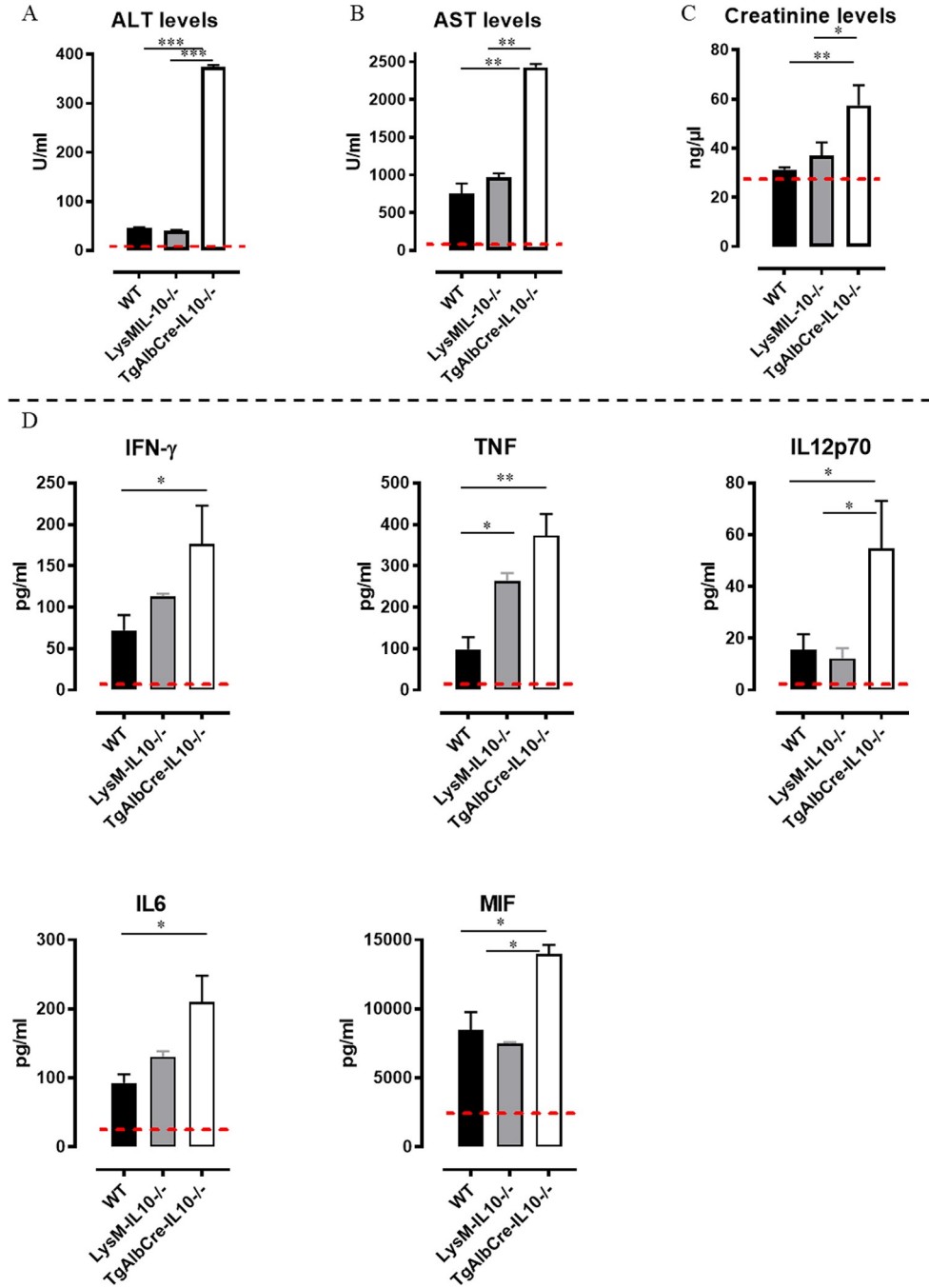

**Fig 5. Hepatocyte-specific IL10-deficiency correlates with an increased tissue pathogenicity and increased serum cytokine levels at day 45 post *T. congolense* infection.** A 45 days p.i. (A) serum ALT, (B) AST and (C) creatinine levels as well as (D) cytokine levels of IFN-γ, TNF, IL12p70, IL-6, MIF and IL-10 were determined via ELISA for wild type (WT, black symbol), LysM-IL-10[-/-] (grey symbol) and TgAlbCre-IL10[-/-] (white symbol) mice. Dashed line represents cytokine levels in non-infected animals. Data are representative of 3 independent experiments and presented as mean of 3 individual mice per group ± SEM. (*: p≤0.05, **: p≤0.01).

(Fig 7C and 7D). Conversely, the total number of RBCs in the blood of infected animals is unaffected compared to naive animals (S5A Fig). Yet, the composition of RBCs (using the gating strategy described in S3B Fig) is altered during infection (S5B Fig), with a decrease in the number of mature RBCs (suggesting enhanced erythrophagocytosis or impaired RBC maturation)

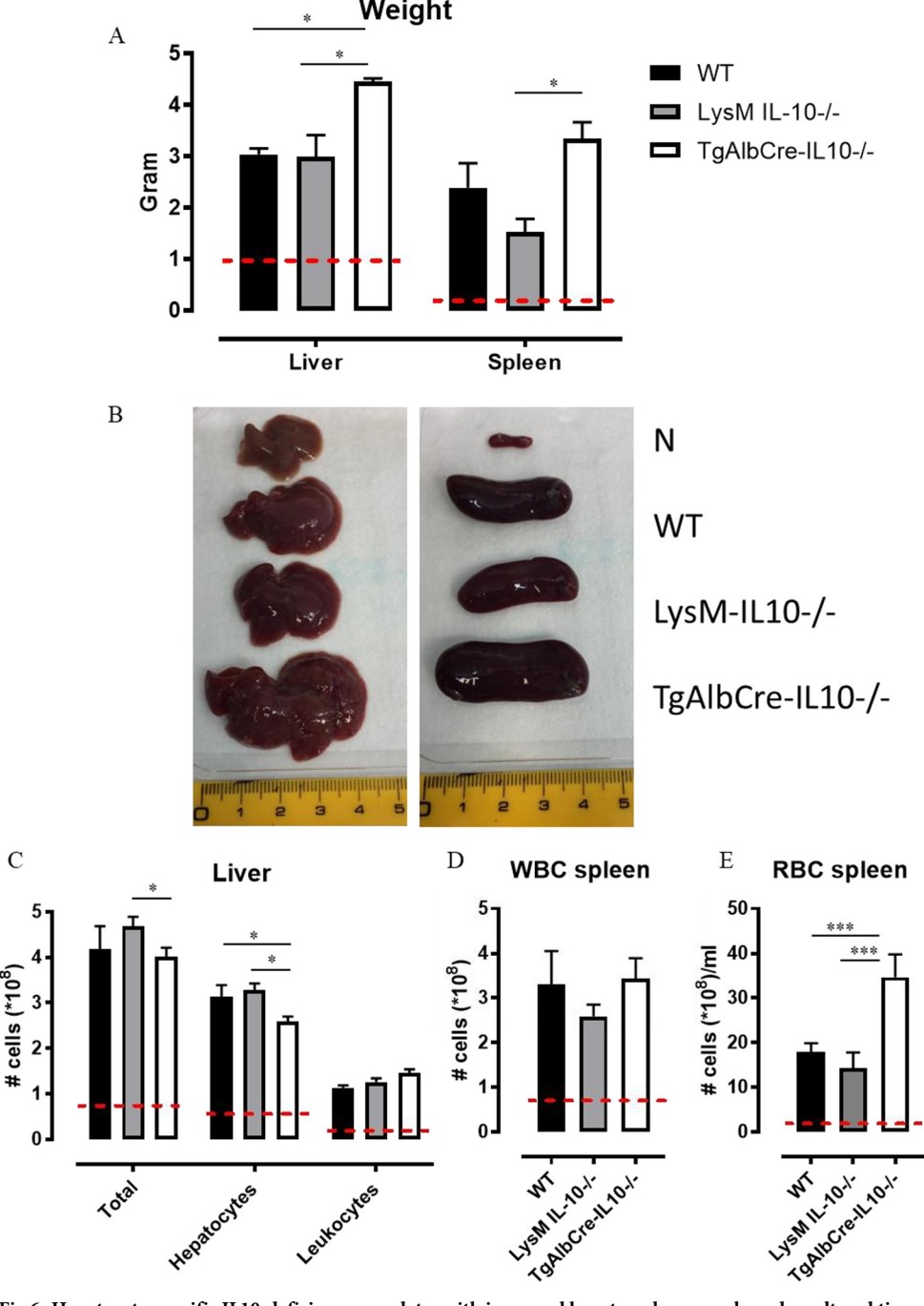

**Fig 6. Hepatocyte-specific IL10-deficiency correlates with increased hepato-splenomegaly and an altered tissue cellular composition at day 45 post *T. congolense* infection.** A) Liver and spleen weight of *T. congolense* infected wild type (WT, black symbol), LysM-IL-10[-/-] (grey symbol) and TgAlbCre-IL10[-/-] (white symbol) mice (days 45 p.i.). (B) Representative images of liver and spleen from naïve and *T. congolense* infected (day 45 p.i.) WT, LysM-IL-10[-/-] and TgAlbCre-IL10[-/-] mice. (C) Total number of liver cells (including hepatocytes and leukocytes), hepatocytes and leukocytes in WT (black symbol), LysM-IL-10[-/-] (grey symbol) and TgAlbCre-IL10[-/-] (white symbol) mice (days 45 p. i.). (D and E) Total number of splenic white blood cells (WBCs) and red blood cells (RBCs) in WT (black symbol), LysM-IL-10[-/-] (grey symbol) and TgAlbCre-IL10[-/-] (white symbol) mice (days 45 p.i.). Dashed line represents cytokine levels in non-infected animals. Data are represented as mean of at least 3–5 mice per group ± SEM and are representative of 2 independent experiments. (*: p≤0.05, ***: p≤0.005).

and an increase in the number of immature RBCs (reflecting enhanced extramedullary erythropoiesis). However, no differences were observed between TgAlbCre-IL10[-/-], LysM-IL-10[-/-] and WT mice in this respect. Enhanced extramedullary erythropoiesis is corroborated by the significantly increased numbers of RBCs in the spleen of infected animals (Fig 6E), encompassing more mature RBCs in TgAlbCre-IL10[-/-] mice compared to the other groups (S5B Fig). These data argue against an impaired RBC maturation in infected mice, and indeed, all RBC maturation stadia (using the gating strategy described in S3B Fig) are similar between WT and TgAlbCre-IL10[-/-] mice (S6 Fig).

In addition, using an *in vivo* erythrophagocytosis assay [31], we observed that TgAlbCre-IL10[-/-] mice exhibited an enhanced erythrophagocytosis both at the level of the spleen and liver compared to WT mice (Fig 8). Hence, the combination of enhanced extramedullary erythropoiesis (resulting in more splenomegaly) and enhanced erythrophagocytosis in TgAlbCre-IL10[-/-] mice might explain the comparable total numbers of RBC in the blood as compared to the other groups.

Collectively, these results indicate that hepatocyte-derived IL-10 plays an important role in attenuating erythrophagocytosis, hemodilution as well as thrombocytopenia.

## Hepatocyte-specific IL10-deficiency correlates with an increased hepatic pro-inflammatory phenotype during *T. congolense* infection

To gain further mechanistic insight in the phenotype of *T. congolense*-infected TgAlbCre-IL10[-/-] mice, we zoomed in on the liver, since the effects of this deficiency are expected to be most apparent within that organ. In first instance, we investigated the local cytokine production by hepatocytes and hepatic leukocytes to assess whether the systemic increase in pro-inflammatory cytokines (Figs 3 and 5D) could be phenocopied at the tissue level in these mice. Quantitative gene expression analysis of the hepatocyte fraction from *T. congolense* infected TgAlbCre-IL10[-/-] mice demonstrated a strongly increased expression of the prototype inflammatory cytokines *Il6* and *Mif* (but not *Tnf*), the inflammatory chemokine *Cxcl10*, and the inflammatory enzyme *Nos2* (but not *Arg1*) compared to hepatocytes from WT and LysM-IL-10[-/-] mice, suggesting an overall more prominent pro-inflammatory profile (Fig 9A). Interestingly, this inflammatory profile is only partially recapitulated in the hepatic CD45[+] leukocyte compartment of infected TgAlbCre-IL10[-/-] mice, with a clear upregulation of *Nos2*, *Cxcl10*, and to a lesser extent *Tnf*, but not *Il6* and *Mif* (Fig 9B). Moreover, *Arg1* is higher expressed in these cells from TgAlbCre-IL10[-/-] mice as compared to WT and LysM-IL-10[-/-] mice, although the *Arg1* mRNA levels remain below those seen in naive animals. Interestingly, CXCL10 is also known as a hepatocyte apoptosis-promoting chemokine [32,33], an important chemokine promoting hepatic inflammation in chronic or acute liver injury through recruitment of leukocytes to the liver parenchyma [34–36]. Hence, its enhanced expression in both hepatocyte- and leukocyte fractions from infected TgAlbCre-IL10[-/-] mice, in association with the lower expression levels of the hepatocyte renewing factor telomerase reverse transcriptase (*Tert1*) in hepatocytes [37] (Fig 9A, lower panels), may collectively explain the lower numbers of hepatocytes in TgAlbCre-IL10[-/-] mice.

At the protein level, hepatocytes from TgAlbCre-IL10[-/-] mice exhibited a significantly increased secretion of IL-6, MIF and nitric oxide (NO; indicative of Nos2 enzyme activity) as compared to hepatocytes from infected WT and LysM-IL-10[-/-] mice, corroborating the mRNA data (Fig 10, upper panels). Hepatic leukocytes from TgAlbCre-IL10[-/-] mice produced the highest levels of NO and IFN-γ (Fig 10, lower panels), while the levels of IL-6, TNF and MIF were increased upon infection but not different between the groups.

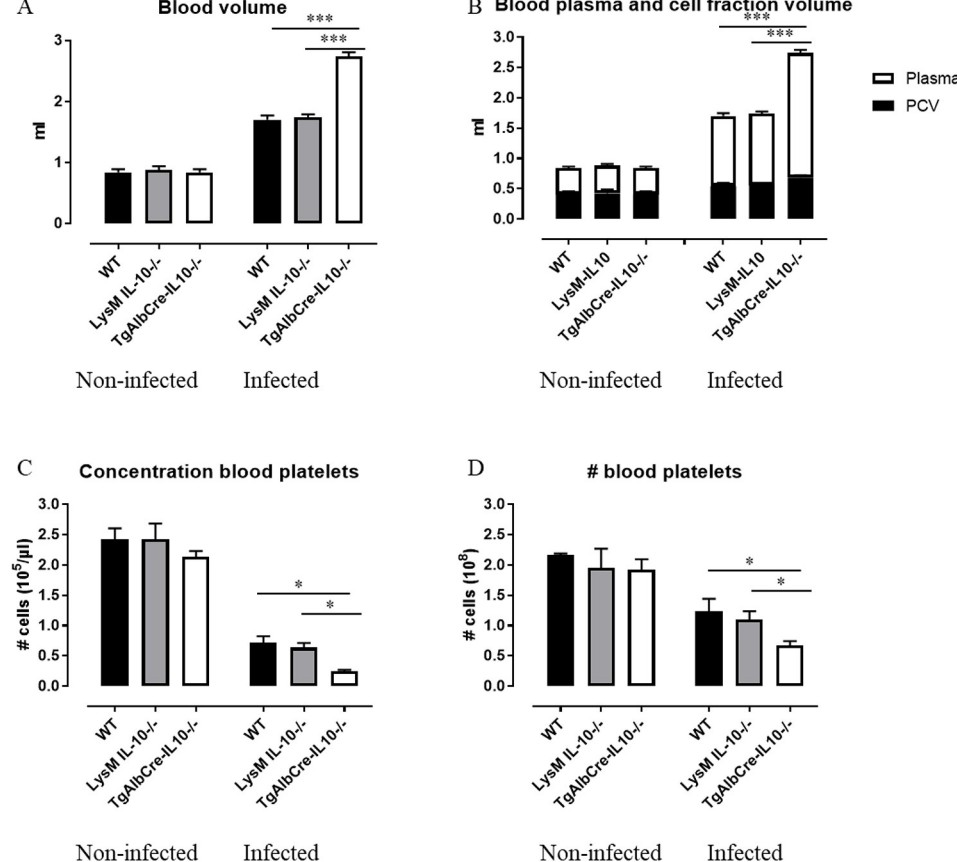

**Fig 7. Hepatocyte-specific IL10-deficiency correlates with increased hemodilution and thrombocytopenia at day 45 post *T. congolense* infection.** Both non-infected and *T. congolense* infected (45 days p.i.) mice were exsanguinated via cardiac puncture and tested for (A) total blood volume and (B) total packed cell volume (PCV, black bar) and total plasma (white bar) volume which were calculated based on the total blood (A) volume and the % PCV. (C) The concentration of CD41[+] platelets (gated as described in [20]) and (D) total number of platelets in the total blood volume was determined was determined for both non-infected and infected animals. WT (black symbol), LysM-IL-10[-/-] (grey symbol) and TgAlbCre-IL10[-/-] (white symbol) mice. Data are represented as mean of at least 3–5 mice per group ± SEM and are representative of 2 independent experiments. (*: p≤0.05, **: p≤0.01, ****: p≤0.001).

Collectively, by comparing WT, LysM-IL-10[-/-] and TgAlbCre-IL10[-/-] mice, hepatocyte-derived IL-10 was found to be crucial to dampen the pro-inflammatory immune response at the level of the liver, decreasing the inflammatory profile of both hepatocytes and leukocytes in this organ. Hereby, hepatocytes from TgAlbCre-IL10[-/-] mice have besides a more pro-inflammatory gene/protein expression profile also an increased *Cxcl10* and decreased *Tert1* expression, which in turn may negatively affect hepatocyte survival.

## Hepatocyte-specific IL10-deficient mice have an enhanced iron-homeostasis gene expression profile during *T. congolense* infection

During chronic inflammation, iron homeostasis is modulated, whereby excess iron can cause hepatocellular damage through the production of harmful reactive oxygen species (ROS) [38]. Hence, also genes involved in iron-regulation and -homeostasis were investigated in both the hepatocyte and hepatic leukocyte fraction and were found to be modulated during *T. congolense* infection (Fig 11). At the level of the hepatocytes (Fig 11, upper panels), these genes were

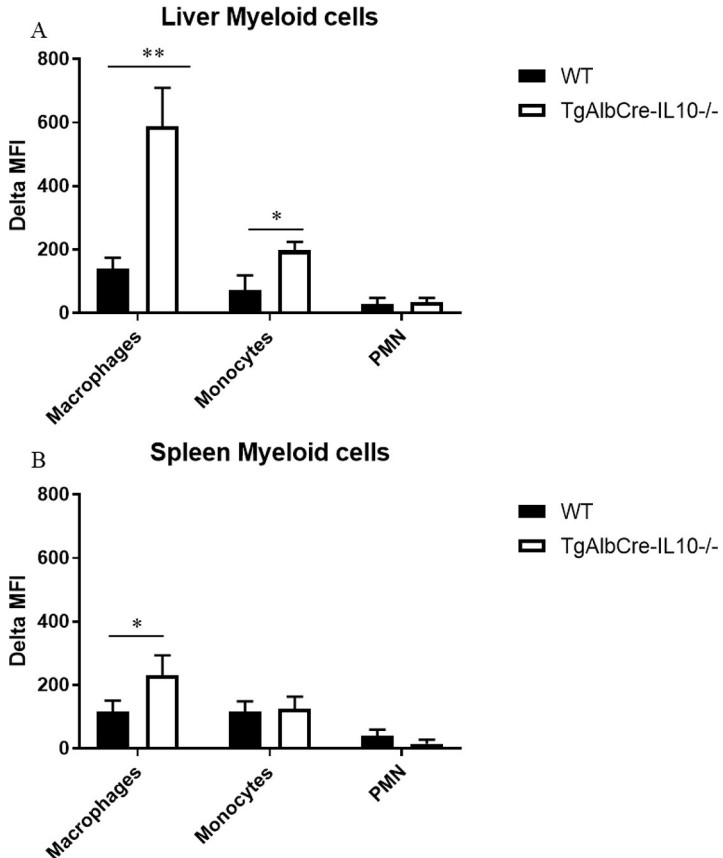

**Fig 8. Hepatocyte-IL10 deficiency enhances erythrophagocytosis during *T. congolense* infection.** At 40 days p.i., $10^9$ pHrodo labelled RBCs isolated from non-infected WT mice were injected i.v. in WT (black bar) or TgAlCre-IL10$^{-/-}$ (open bar) mice. 18 h later, mice were sacrificed and liver (A) and spleen (B) myeloid phagocytic cells (MPC), namely CD11b$^+$Ly6C$^{int}$Ly6G$^+$ PMNs, CD11b$^+$Ly6C$^{high}$Ly6G$^-$ monocytes and CD11b$^+$Ly6C$^-$Ly6G$^-$F4/80$^+$ macrophages (identified as described in [20]) were tested for delta median fluorescence intensity (MFI) of the intracellular pHrodo signal determined by subtracting the PE signal of cells from mice receiving unlabeled RBCs from the PE signal of cells from mice receiving pHrodo-labeled RBCs. Results are representative of 2 independent experiments and presented as mean of 3 individual mice ± SEM. $^*$: p≤0.05, $^{**}$: p≤0.01.

skewed towards an increased iron uptake (*i.e.* higher *Hmox-1*, *Nramp2*) and retention (*i.e.* higher *Fhc*, *Lcn-2*) in TgAlbCre-IL10$^{-/-}$ mice. Accordingly, the gene expression profile of iron regulating genes in liver leukocytes is also suggestive of an amplified iron uptake (*i.e.* higher *Hmox-1*, *Lcn-2*) and iron processing (*i.e.* higher *Fpn-1*, *Fhc*) in TgAlbCre-IL10$^{-/-}$ mice compared to the other two groups (Fig 11, lower panels).

Collectively, the hepatocyte and hepatic leukocyte iron-homeostasis gene analysis of TgAlbCre-IL10$^{-/-}$ mice are suggestive of an enhanced iron uptake and retention, which in turn may favour hepatic damage.

## Anti-MIF Nb treatment attenuates effects of Hepatocyte-specific IL10-deficiency during *T. congolense* infection

Finally, we wished to ascertain whether the increased inflammatory cytokine production is responsible for the increased pathogenicity of *T. congolense* infection in TgAlbCre-IL10$^{-/-}$ mice. In this respect, we have reported before that MIF plays a key role in *T. congolense* infection associated pathogenicity [39] and we demonstrated here that MIF levels were significantly upregulated in TgAlbCre-IL10$^{-/-}$ mice during later stages of infection both systemically and in

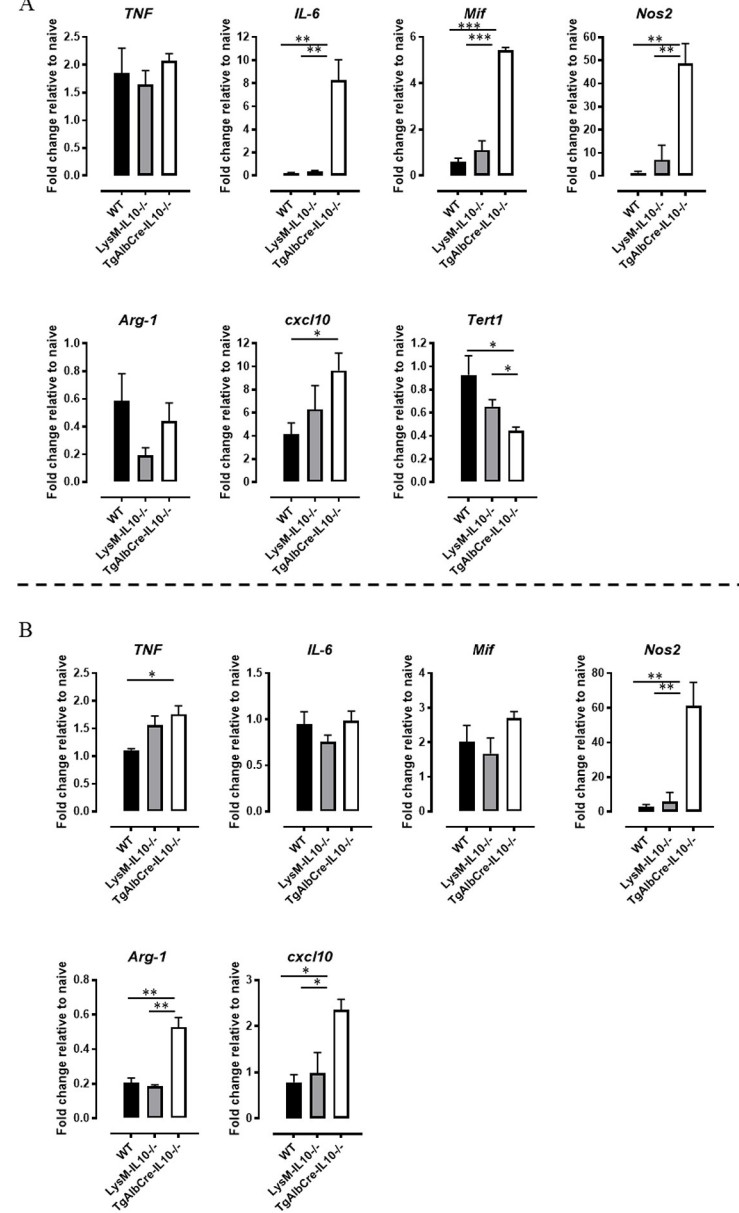

**Fig 9. Gene expression profiles of pro- and anti-inflammatory cytokines/mediators as well as chemokines and *Tert-1* in hepatocytes and liver-associated leukocytes from *T. congolense* infected animals.** Gene expression levels of pro-inflammatory and anti-inflammatory cytokines/mediators as well as chemokines in hepatocytes (A, upper panel) and liver leukocytes (B, lower panels) fraction of *T. congolense* infected (day 45 p.i.) WT (black symbol), LysM-IL-10[−/−] (grey symbol) and TgAlbCre-IL10[−/−] (white symbol) mice. Results are presented as fold change whereby the expression levels were normalized using *S12* and expressed relatively to the expression levels in the corresponding non-infected animals. Gene expression levels of *Tert1* in the hepatocyte fraction of WT (black symbol), LysM-IL-10[−/−] (grey symbol) and TgAlbCre-IL10[−/−] (white symbol) mice. Data are represented as mean of at least 3–5 mice per group ± SEM and are representative of 2 independent experiments. (*: p≤0.05, **: p≤0.01, ***: p≤0.005).

the liver (Figs 3 and 10A). Moreover, in contrast to IL-6, MIF is an upstream regulator of the inflammatory cascade whose blockade may be more efficient at attenuating the inflammatory immune response [40]. Hence, we evaluated whether neutralization of MIF, using a half-life extended anti-MIF Nb [39], could attenuate the pathology observed in TgAlbCre-IL10[−/−] mice. To this end, TgAlbCre-IL10[−/−] mice were treated starting from day 30 p.i. (*i.e.* a time point

when anemia difference between the groups became apparent, see Fig 3D) with 250 μg of a half-life extended anti-MIF Nb for ~3 weeks (twice per week) and infection parameters (anemia and parasitemia) were monitored. The anti-MIF Nb treatment did not affect parasitemia development in TgAlbCre-IL10$^{-/-}$ mice (Fig 12A, upper panel), but it clearly prevented anemia from further progressing (Fig 12A, lower panel). Since inflammatory cytokines play a key role in anemia development and since MIF can be an initiator of inflammatory cytokine production, we also evaluated the systemic levels of various pro-inflammatory cytokines. Interestingly, treatment of TgAlbCre-IL10$^{-/-}$ mice with an anti-MIF Nb resulted in a significant reduction in systemic TNF, IL-6 and MIF levels and this to levels comparable to those observed in WT mice (Fig 12B).

Collectively, blocking MIF in TgAlbCre-IL10$^{-/-}$ mice, using an anti-MIF Nb, was found to attenuate anemia and excessive systemic inflammatory cytokine levels, without affecting parasite burden.

## Discussion

Trypanosomatids are very proficient in evading the host's immune response, yet, these escape mechanisms are associated with a chronic inflammatory state that culminates into increased tissue damage and finally into host death [41]. IL-10 was shown to be essential to attenuate the inflammatory response and prevent early death of the host due to a hyper-inflammation syndrome [19]. Hence, a well-timed balance between the levels of pro-inflammatory cytokines (required for initial parasite control) and anti-inflammatory cytokines (required to dampen

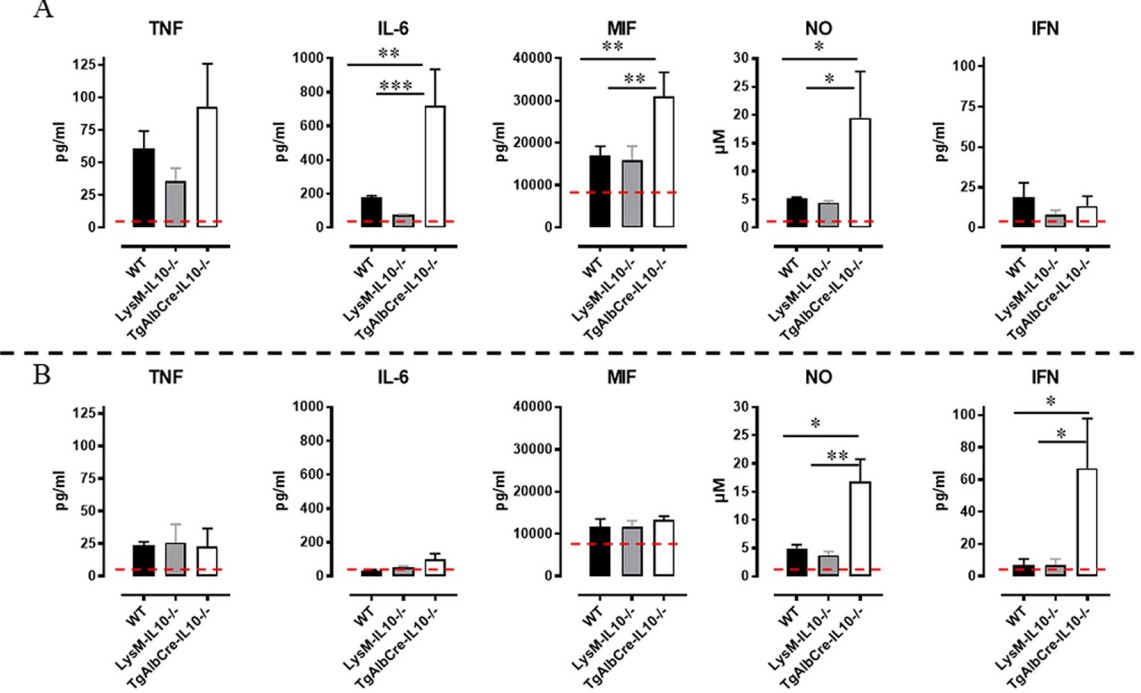

**Fig 10. Cytokine profile of liver cell cultures following *T. congolense* infection.** At day 45 post *T. congolense* infection, fractionated liver cells (hepatocytes and leukocytes) were isolated and cultured for 36 hours and subsequently tested in ELISA for cytokine levels of IFN-γ, TNF, IL-6 and MIF as well as NO. Cytokine protein levels of hepatocytes (A, upper panels) and liver leukocytes (B, lower panels). WT (black symbol), LysM-IL-10$^{-/-}$ (grey symbol) and TgAlbCre-IL10$^{-/-}$ (white symbol) mice. Dashed line represents cytokine levels in non-infected animals. Non-infected animals did not show any detectable cytokine levels. Data are represented as mean of at least 3–5 mice per group ± SEM and are representative of 2 independent experiments. (*: p≤0.05, **: p≤0.01, ***: p≤0.005).

excessive pro-inflammatory effects) is essential to prevent excessive tissue damage. Previous research has shown that leukocytes (*i.e.* Tregs. alternatively activate macrophages and Ly6C⁻ patrolling monocytes) can constitute a source of IL-10 required to dampen some specific aspects of immunopathology (*e.g.* liver injury) during the early stages of infection [14,15,19,24]. Using hepatocyte-specific IL-10 deficient (TgAlbCre-IL10⁻/⁻) mice and IL-10-eGFP reporter mice we now show for the first time that hepatocytes from *T. congolense* infected mice can also produce IL-10 (see Fig 1C). Moreover, we provide evidence that parasite-derived components, can trigger IL-10 production by hepatocytes from non-infected animals, indicating that they can sense and respond to the presence of trypanosomes. In this context, in another parasitic model, it was recently shown that hepatocytes can respond to *Leishmania infantum* infection by the activation of inflammatory mechanisms and the production of IL-10 in order to balance the inflammatory response and avoid cell damage [42].

The potential of hepatocytes to produce IL-10 was found to be crucial to control immuno-pathology during the chronic stage of *T. congolense* infection. Indeed, although TgAlbCre-IL10⁻/⁻ mice behaved similar to WT and LysMCre-IL10⁻/⁻ mice during the early stages of infection, they succumbed much earlier to the infection coinciding with increased tissue path-ogenicity. This was evidenced by the increased ALT (suggesting liver injury), AST (suggesting systemic injury) and creatinine (suggesting kidney failure) levels, as well as by a more severe chronic anemia. The enhanced pathogenicity correlated with a significant systemic increase in all pro-inflammatory cytokines documented to play a pathogenic role during *T. congolense* infection [20,27,43]. In addition, at later stages of infection, the numbers of PMN as well as the expression of MHC-II on Ly6C⁺ inflammatory monocytes, both documented to contribute to tissue injury [22,26], were found to be significantly increased in the blood of TgAlbCre-IL10⁻/⁻ mice compared to the other groups, further strengthening the notion that these mice were in a higher inflammatory state. In contrast, the numbers of Ly6C⁻ patrolling monocytes,

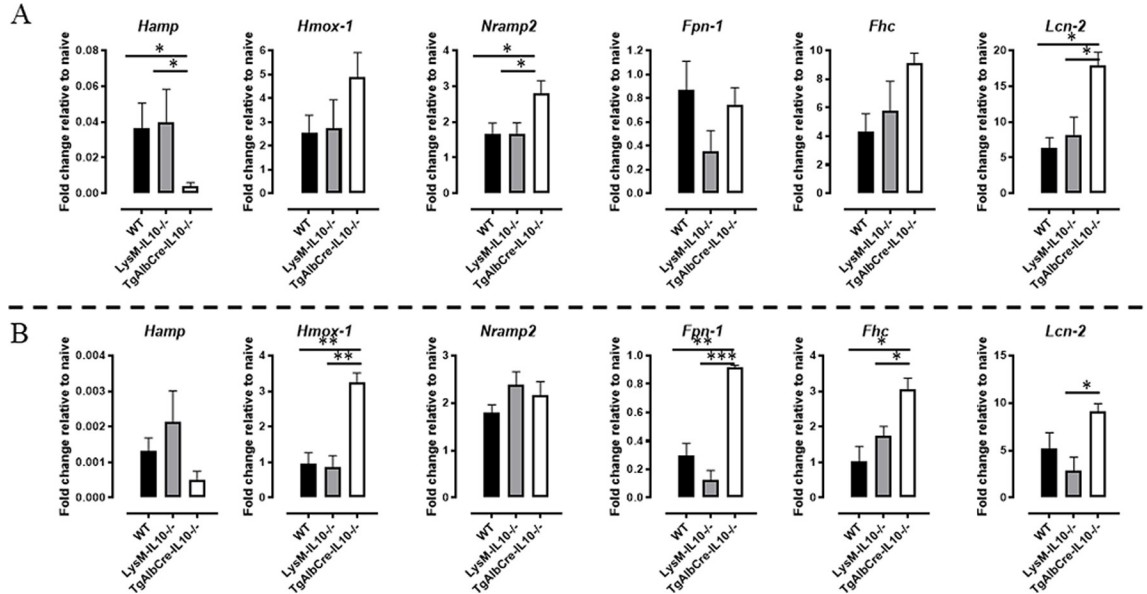

**Fig 11. Iron-homeostasis associated gene expression profiles of hepatocytes and liver-associated leukocytes from *T. congolense* infected animals.** Gene expression levels of iron-homeostasis associated genes in hepatocytes (A, upper panels) and the liver associated leukocyte (B, lower panels) fraction of WT (black symbol), LysM-IL-10⁻/⁻ (grey symbol) and TgAlbCre-IL10⁻/⁻ (white symbol) mice. Results are presented as fold change whereby the expression levels were normalized using *S12* and expressed relatively to the expression levels in the corresponding non-infected animals. Data are represented as mean of at least 3–5 mice per group ± SEM and are representative of 2 independent experiments. (*: p≤0.05, **: p≤0.01, ***: p≤0.005).

documented to be important to attenuate tissue damage caused by an extensive Ly6C[+] mono-cyte-associated inflammatory immune response [24], were found to be significantly reduced in TgAlbCre-IL10[-/-] mice compared to WT and LysM-IL-10[-/-] mice. Furthermore, the number of circulating B cells was found to be similar in both groups of mice and was only significantly reduced in TgAlbCre-IL10[-/-] mice at later stages of infection. Since antibodies are required to keep the parasitemia in check [11,44], the reduced B cell presence in the blood could in turn explain the higher parasitemia levels observed in TgAlbCre-IL10[-/-] mice at later stages of infection. It is however important to realize that B cell counts in the blood do not reveal whether B cell activation, the formation of plasma cells and antibody secretion, the induction of germinal center reactions, or the activation of CD4[+] T follicular helper cell numbers and function, which are known to restrict *T. congolense* parasitemia [11,44], are impaired. However, this is highly likely since trypanosomes cause a general B cell depletion pathology, which is initiated by the very rapid disappearance of immature B cells in the bone marrow, as well as transitional and IgM[+] marginal zone B cells from the spleen, followed by a gradual depletion of Follicular B cells (FoB) [45,46]. Though the cause of B cell depletion is currently unknown, it was shown that the kinetics of this B cell loss and parasite outgrowth depends on different factors such as parasite virulence and the level of inflammation [45,46]. Therefore, the observed reduced B cell numbers in TgAlbCre-IL10[-/-] mice at the later phase of infection is most likely due to a higher/persistent inflammatory immune response [45,47]. In turn, this higher and persistent inflammatory immune response can also contribute to multiple organ failure and host death [19,23]. Hence, during the chronic stage of the infection, hepatocyte-derived IL-10 seems to be more crucial to dampen the inflammatory immune response than leukocyte-derived IL-10. A more in-depth analysis of the liver at this stage of infection revealed that TgAlbCre-IL10[-/-] mice had less hepatocytes, but the remaining hepatocytes showed a strongly increased pro-inflammatory gene expression and protein secretion as compared to WT or LysMCre-IL10[-/-] mice. Moreover, the protein and gene expression levels of *IL-6* and *Mif*, two cytokines documented to play a key role in *T. congolense*-associated pathology (*i.e.* susceptibility versus tolerance) and liver injury in particular [20,48–50], were drastically increased in hepatocytes from TgAlbCre-IL10[-/-] mice compared to the other groups. Interestingly, we have shown before that MIF is a key player in *T. congolense*-associated pathogenicity [20]. Indeed, absence of the pro-inflammatory regulator MIF, using *Mif*-deficient mice, was shown to attenuate the inflammatory immune response during the chronic stage of infection resulting in a reduced infection-associated pathogenicity (*i.e.* anemia, liver and tissue injury) and prolonged survival by promoting IL-10 production [20,22]. Notably, hepatocyte-derived MIF was also shown in another model (*i.e.* alcoholic liver disease) to be a driving force for liver injury [50]. Overall, an interesting picture emerges whereby hepatocytes are a source of two cytokines that counterbalance each other and exert opposing effects during chronic *T. congolense* infection: MIF attenuates IL-10 production and promotes pathogenicity; while IL-10 keeps the MIF production in check and softens pathogenicity. Hence, therapeutic approaches that tilt the balance of "hepatocyte" cytokine production in favour of IL-10 are likely to majorly impact the wellbeing and survival of *T. congolense*-infected animals. Given that, in contrast to IL-6, MIF is an upstream regulator of the inflammatory cascade whose blockade may be more efficient at attenuating the inflammatory immune response [40]. This notion was strengthened by the fact that interfering with MIF signaling using a Nb-based approach indeed attenuates the excessive inflammatory immune response as well as the most prominent pathological feature associated with *T. congolense* infections, *i.e.* anemia, in TgAlbCre-IL10[-/-] mice.

Our results further strengthen the notion that during the course of *T. congolense* infection, a sequential transition occurs in the cells that produce IL-10. During the early stages, leukocytes seem to play a decisive role as IL-10 producing cells, yet during the later stage hepatocytes

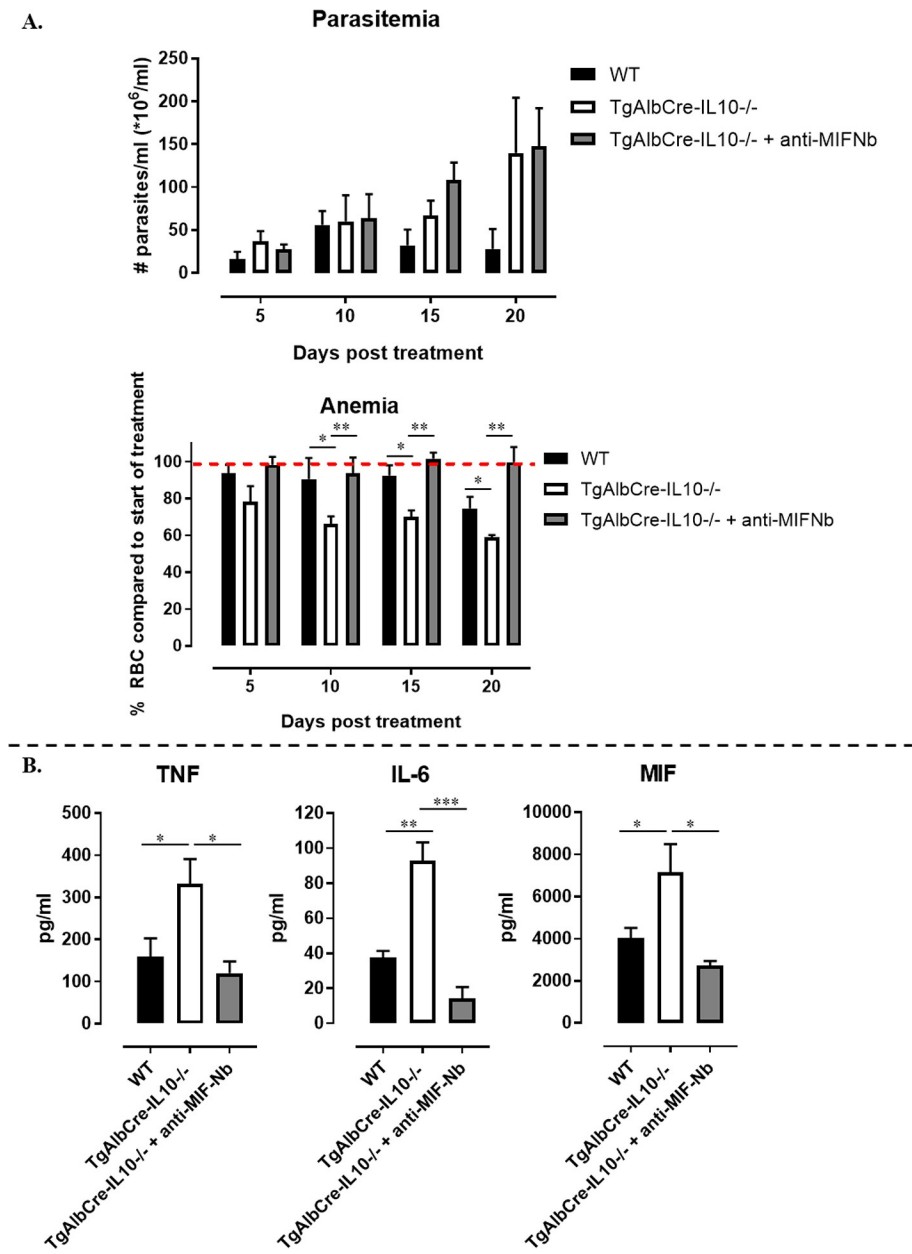

**Fig 12. Half-life extended anti-MIF Nb treatment attenuates anemia and pro-inflammatory cytokine levels in Hepatocyte-IL10 deficient mice during *T. congolense* infection.** At 30 days p.i., TgAlbCre-IL10$^{-/-}$ mice (4–5 mice/group) were treated i.p. with 250 µg half-life extended anti-MIF Nbs (grey bars) or left untreated (white bars) for 3 weeks (twice a week). Wild type mice (black bars) were left untreated and included as negative controls. A. Parasitemia (upper panel) and anemia (lower panel) progression expressed in percentage relative to levels recorded prior to initiating the treatment were monitored for 3 weeks. B. Serum cytokine levels recorded 3 weeks post treatment. Data are represented as mean of at least 4–5 mice per group ± SEM. (*: p≤0.05, **: p≤0.01, ***: p≤0.005).

become important IL-10 producing cells in an attempt to counteract the host's inflammatory response. As a matter of fact, at this later stage of *T. congolense* infection, hepatocyte-derived IL-10 plays a decisive role to dampen excessive tissue damage and immunopathology. More-over, hepatocytes from WT infected mice, but not TgAlbCre-IL10$^{-/-}$ infected mice, exhibited an increased *IL-10R* gene expression, suggesting that autocrine/paracrine IL-10R signalling on hepatocytes may be required to allow efficient control over inflammation. Additional

experimentation will be required to prove this point. One of the important features that is regulated by hepatocyte-derived IL-10 is the expression of iron homeostasis-associated genes, suggesting an enhanced liver iron-uptake/retention in TgAlbCre-IL10$^{-/-}$ mice. Indeed, TgAlbCre-IL10$^{-/-}$ mice exhibited an increased expression of *Hmox-1* (indicating enhanced iron uptake) as well as *Fpn-1* (indicating enhanced iron export) and *Fhc* (indicating enhanced iron retention). These modulations in iron-homeostasis regulating genes are suggestive for an enhanced erythrophagocytosis and a reduced iron availability for erythropoiesis [51], which was also reflected by the lower number of mature RBCs in the blood (S5A Fig). In this context, we confirmed that TgAlbCre-IL10$^{-/-}$ mice exhibited an enhanced erythrophagocytosis, yet, the erythropoiesis efficiency was not affected. This strengthens the notion that iron is retained into liver cells in the absence of hepatocyte IL-10, which may then further fuel oxidative stress and liver injury [52]. Moreover, the increased hepatosplenomegaly in TgAlbCre-IL10$^{-/-}$ mice leads to hemodilution, which in combination with a reduced erythropoiesis aggravates one of the most prominent immunopathological features of *T. congolense*-infected animals: anemia. In this context, MIF might again be an important player as it was shown previously to be a driving force in suppressing erythropoiesis and promoting hemodilution leading to the observed "apparent" anemia during *T. congolense* infection [20].

Overall, the crucial role of hepatocytes at the chronic stage of infection might be linked to their inherent ability to contribute to liver regeneration [37]. Given that hepatocytes play a key role in liver homeostasis, identification of target molecules or pathways that are modulated between trypano-susceptible and -tolerant animals might open perspectives to develop a more specific approach to alleviate infection-associated pathogenicity and allow a restoration of normal organ functions. Since MIF was shown to be a key player in trypanosomosis-associated pathogenicity and found to be strongly induced in hepatocytes from TgAlbCre-IL10$^{-/-}$ mice compared to the other groups, and interfering with MIF signaling using a Nb-based approach attenuates the excessive inflammatory immune response as well as the most prominent pathological feature associated with *T. congolense* infections, *i.e.* anemia, in TgAlbCre-IL10$^{-/-}$ mice, this could be a prime candidate for further studies in this model. Therefore, using MIF blocking Nbs at later stages of infection might be a novel way to attenuate *T. congolense* associated pathology and allow animals to remain productive.

# Materials and methods

## Ethics statement

All experiments, maintenance and care of the mice complied with the European Convention for the Protection of Vertebrate Animals used for Experimental and Other Scientific Purposes guidelines (CETS n˚ 123) and were approved by the Ethical Committee for Animal Experiments (ECAE) at the Vrije Universiteit Brussel (Permit Numbers: 14-220-05 and 14-220-06).

## Parasites, mice and infections

Eight to twelve weeks old female C57BL/6 mice were purchased from Janvier, France. IL-10$^{-/-}$ (B6.129P2-*Il10*$^{tm1Cgn}$/J), *AlbCre* (B6.Cg-Tg(Alb-cre)21Mgn/J, JAX stock #003574), *LysMCre* (B6.129P2-*Lyz2*$^{tm1(cre)Ifo}$/J, JAX stock #004781) and Vert-X (B6(Cg)-*Il10*$^{tm1.1Karp}$/J) were purchased from Jackson Laboratory, USA. *LyzM*$^{Cre/+}$ IL-10$^{fl/fl}$ (*i.e.* LysM-IL10) and *Alb*$^{Cre/+}$ IL-10$^{fl/fl}$ (*i.e.* TgAlbCre-IL10) mice were generated in house by crossing the *LysMCre* and *AlbCre* mice with *IL-10*$^{fl/fl}$ mice (a kind gift of W. Muller, University of Manchester, Manchester, United Kingdom), respectively. The genotyping profile of the LysM-IL10$^{-/-}$ and TgAlbCre-IL10$^{-/-}$ mice is included (S7 Fig).

Clonal *T. congolense* parasites (Tc13) were kindly provided by Dr. Henry Tabel (University of Saskatchewan, Saskatoon) and stored at -80˚C. Female mice (7–8 weeks old) were infected intraperitoneally (i.p.) with $2 \times 10^3$ Tc13 trypanosomes. Parasite and red blood cell (RBC) numbers in blood were determined via hemocytometer by tail-cut (2.5 µl blood in 500 µl RPMI). Anemia was expressed as the percentage of reduction in RBC counts compared to non-infected animals. Packed cell volume (PCV) was measured following collection of anti-coagulated blood in heparinized capillaries and centrifugation at 9500×g for 7 min. using a micro-centrifuge (Fisher BioBlock Scientific). Also the weight of the mice was followed during the course of infection and the percentage weight change compared to non-infected animals at the corresponding time.

Treatment experiments consisted of injecting TgAlbCre-IL10$^{-/-}$ mice (4–5 mice/group) i.p. at day 30 post infection (p.i.) with 250 µg (in 200 µl PBS) of a half-life extended anti-MIF Nb [39] and this for 3 weeks (twice a week). At regular time points blood was collected for parasitemia and RBC analysis and when the untreated TgAlbCre-IL10$^{-/-}$ mice started dying blood was collected for cytokine analysis. Of note, anemia levels at day 30 p.i. of each individual mouse were used to normalize the progression of anemia.

## Serum and cell isolation

Blood from non-infected control and infected mice was harvested via tail-cut using heparinized capillaries and centrifuged at 8000×g for 15 min. Serum was harvested and stored at -20˚C. Hepatocytes and liver leukocytes from infected and non-infected mice were purified as described in [53]. Briefly, liver cells were obtained by perfusing the liver with 10 ml of cold PBS via the inferior vena cava, mechanical disruption of the liver, followed by passing cell suspensions over a 70 µm nylon mesh filter and a 33% Percoll gradient separation, after which cells were resuspended in complete medium (RPMI supplemented with 5% FCS, 2mM L-glutamine, 100 U/ml penicillin and 100 µg/ml streptomycin (all from Invitrogen Life Technologies)). Next, the fractionated cell suspension was centrifuged (7 min., 300×g, 4˚C) and the pellet treated with erythrocyte-lysis buffer. Following centrifugation (7 min., 300×g, 4˚C) the pellet was resuspended in 2–5 ml complete medium, cells counted and adjusted at $10^7$ cells/ml for flow-cytometric analysis, cell culturing and RT-PCR analysis. Of note, the 33% Percoll gradient yielded hepatocytes (in the upper layer) with at least 95% purity (see Fig 1A and 1B).

Spleen cells from infected and non-infected mice were obtained by homogenizing (disrupted mechanically) the organs in 10 ml RPMI/5%FCS medium, passing the suspension through a 70 µm nylon mesh filter and centrifugation (7 min., 300×g, 4˚C). Cells were counted and brought at $10^7$ cells/ml in RPMI/5% FCS medium for RBC analysis via flow cytometry. Remaining cells were pelleted (7 min., 300×g, 4˚C) and subsequently treated with RBC lysis buffer (0.15 M NH$_4$Cl, 1.0 mM KHCO$_3$, 0.1 mM Na$_2$-EDTA) and processed as described for the liver (see above) for analysis of white blood cells (WBCs).

## Flow cytometry

To analyze the RBC composition, the blood and spleen cells were analysed omitting RBC lysis. Briefly, total blood (2.5 µl diluted in 500 µl RPMI/5% FCS) and $10^6$ spleen cells (in 100 µl) were incubated (15 min., 4˚C) with Fc-gamma blocking antibody (anti-CD16/32, clone 2.4G2, BD Biosciences), and subsequently stained with labelled antibodies (summarized in S1 Table) or matching control antibodies. Samples were washed with FACS medium (5% FCS in HBS), measured on FACSCanto II flow cytometer (BD Bioscience) and data were analysed using FlowJo software (Tree Star Inc., Ashland, OR) by excluding CD45$^+$ cells and gating on Ter-119$^+$ cells. After RBC lysis, the remaining cells ($10^6$ cells/100 µl) within the spleen (comprising

leukocytes, WBCs) and liver (comprising leukocytes and hepatocytes) were analyzed as described above. The results were analysed after exclusion of aggregated and death cells (7AAD⁺, BD Pharmingen) and selection of CD45⁺ cells (leukocytes) and CD45⁻ cells (hepatocytes). The total number of cells in each population was determined by multiplying the percentages of subsets within a series of marker negative or positive gates by the total cell number determined for each tissue.

## Erythrophagocytosis assay

The pHrodo-labelling of red blood cells (RBCs) was described in [31]. $10^9$ pHrodo-labelled RBCs isolated from non-infected wild type (WT) mice were injected i.v. in WT or TgAlbCre-IL-10$^{-/-}$ mice. 18 h later, mice were sacrificed, and liver and spleen cells isolated as described above. Next, via FACS (see S1 Table), liver and spleen CD11b$^+$Ly6C$^{int}$Ly6G$^+$ PMNs, CD11b$^+$Ly6C$^{high}$Ly6G$^-$ monocytes and CD11b$^+$Ly6C$^-$Ly6G$^-$F4/80$^+$ macrophages were tested for delta median fluorescent intensity (MFI) of the intracellular pHrodo signal determined by subtracting the PE signal of cells from mice receiving unlabelled RBCs from the PE signal of cells from mice receiving pHrodo-labelled RBCs.

## *In vitro* cultures

Liver leukocytes and splenic cell populations isolated from infected and non-infected mice were diluted in complete medium (RPMI-1640 medium, 5% FBS, 1% sodium pyruvate (Gibco), 1% non-essential amino acids (Gibco), 1% glutamate, 1% penicillin-streptomycin), plated in 48-well plates (Nunc) at $2.10^6$ cells per ml and incubated at 37˚C in a 5% $CO_2$ incubator for 36–48 h before supernatant was recovered and stored at -20˚C for further analysis. The isolated hepatocytes were cultured in DMEM (Thermo-Scientific), supplemented with 5% FBS and 100 mg ml$^{-1}$ penicillin/streptomycin, at $2.10^6$ cells per ml and incubated at 37˚C in a 5% $CO_2$ incubator for 36–48 h before supernatant was recovered and stored at -20˚C for further analysis.

For the *in vitro* co-cultures of hepatocytes and liver leukocytes with parasites or parasite lysate or parasite derived sVSG, parasites were isolated as described in [54,55]. Briefly, mice with a systemic *T. congolense* (Tc13) parasitemia were exsanguinated and parasites were purified from heparinized blood by DEAE-cellulose (DE-52, Whatman) chromatography and sVSG was isolated via ion-exchange chromatography and gel filtration. Lysate was prepared by 3 repetitive freeze thawing cycles (-80˚C, 37˚C). The concentration of both sVSG and lysate was determined spectrophotometrically (Nanodrop) following Prosep-Remtox (Immunosource, Schilde, Belgium) treatment and samples confirmed to be LPS free by using the Limulus Amebocyte Lysate Kinetic-QCL Kit (Cambrex, East Rutherford, NJ, USA) in accordance with the manufacturer's instructions. Both sVSG and lysate were either used immediately or stored at −20˚C. Finally, $2.10^6$ liver cells per ml were incubated at 37˚C in a 5% $CO_2$ incubator with $10^7$ parasites or 50 µg lysate or 5 µg sVSG or left untreated (negative control) for 36 hours after which the culture supernatant was collected and stored at -80˚C for further cytokine ELISA analysis.

## Quantification of cytokines

All serum cytokines except MIF were quantified using the V-PLEX Custom Mouse Cytokine kit (catalogue number K152A0H) from Meso Scale Discovery (Rockville, MD, USA) according to the manufacturer's protocol. The MIF protein was quantified using a kit from R&D Systems according to the manufacturers description. Alternatively, culture medium concentrations of

MIF, TNF, IFN-γ and IL-10 (R&D Systems) as well as IL-6 (Pharmingen) were determined by ELISA as recommended by the suppliers.

## Real-time quantitative polymerase chain reaction (RT-QPCR) analysis

One μg of total RNA prepared from $10^7$ cells (RNeasy plus mini kit, Qiagen) was reverse transcribed using oligo(dT) and Superscript II Reverse Transcription following the manufacturer's recommendations (Roche Molecular Systems). RT-QPCR was performed in an iCycler iQ, with iQ SYBR Green Supermix (Bio-Rad) as described in [56]. Primer sequences are listed in S2 Table. PCR cycles consisted of 1-minute denaturation at 94˚C, 45-second annealing at 55˚C, and 1-minute extension at 72˚C. Fold change in gene expression was expressed as compared to non-infected animals after normalization against the Ct value of the ribosomal *S12* (*Mrps12*) protein as household gene.

## Aspartate transaminase (AST), alanine transaminase (ALT) and creatinine measurement

Serum AST and ALT levels were determined as described by the suppliers (Boehringer Mannheim Diagnostics). Serum creatinine levels were determined as described by the suppliers (Abcam).

## Statistics

The GraphPad Prism 7 software was used for statistical analyses. For samples that complied with the D'Agostino-Pearson test, a student *t*-test for paired analyses or one-way ANOVA with multiple comparisons was performed and a Log-rank (Mantel-Cox) test was used for survival. Alternatively, a non-parametric Mann-Whitney (two groups) and Kruskal-Wallis (three or more groups) was performed for experiments that did not comply with the D'Agostino-Pearson test. Values are expressed as mean ± SEM. Values of p≤0.05 are considered statistically significant, where * = p ≤ 0.05, ** = p ≤ 0.01 and *** = p ≤ 0.001.

## Supporting information

**S1 Table. Fluorescently labeled antibodies used.**
(DOCX)

**S2 Table. Primer used for RT-PCR analysis.**
(DOCX)

**S1 Fig. Leukocytes are proficient IL-10 producing cells during the chronic phase of *T. congolense* infection.** Representative FACS profile of purified leukocytes at day 45 post *T. congolense* infection. The gating strategy used to discriminate between leukocytes and potential contaminating hepatocytes/debris is based on an FSC-A versus SSC-A (A) and a CD45 versus FSC-A (B) plot, whereby leukocytes were selected based on their CD45⁺ profile. (C) Histogram plot showing the intensity of the IL-10-eGFP signal in leukocytes from IL-10-eGFP reporter (blue) mice and, as negative control, in leukocytes from TgAlbCre-IL10⁻/⁻ (red) mice. At day 45 post *T. congolense* infection, isolated hepatocytes from WT (black symbol) and TgAlbCre-IL10⁻/⁻ (white symbol) mice were cultured for 36 hours and subsequently tested in ELISA for IL-10 protein levels (D) or tested in RT-PCR for *IL-10* and *IL-10R* gene expression (E and F, respectively). Of note, RT-PCR results are presented as fold change whereby the expression levels were normalized using *S12* and expressed relatively to the expression levels in the corresponding non-infected animals. Non-infected animals did not show any detectable IL-10 protein levels (Dashed line). Data are

represented as mean of at least 3–5 mice per group ± SEM and are representative of 2 independent experiments. (*: p≤0.05, **: p≤0.01, ***: p≤0.005). ND: Not detected.
(TIF)

**S2 Fig. Hepatocyte-specific IL10-deficiency does not affect survival and tissue pathogenicity during *T. brucei* infection.** A) Parasitemia, (B) Survival, (C) weight change, (D) anemia of *T. brucei* infected wild type (WT, black symbol) and TgAlbCre-IL10$^{-/-}$ (red symbol) mice. Data are represented as mean (A, C-G) or median (B) of 3–5 mice per group ± SEM and are representative of 2–3 independent experiments.
(TIF)

**S3 Fig. Gating strategy used to identify different leukocytes and red blood cells (RBCs) within the blood during *T. congolense* infection.** Representative FACS profiles on blood of *T. congolense* infected animals to identify different leukocyte (A) and RBC (B) subsets. For the leukocyte, first, a CD45 versus FSC-A plot allows identifying CD45$^+$ cells, after which these cells put in a CD11b versus FSC plot to identify CD11b$^+$ cells and CD11b$^-$ cells (lymphocytes). The CD11b$^-$ cells (lymphocytes) were then plot in a CD19 versus MHC-II plot to identify B cells (CD19$^+$MHC-II$^+$) and T cells (CD19$^-$MHC-II$^-$). The CD11b$^+$ cells were plotted in a Ly6C versus Ly6G plot to identify inflammatory monocytes (Ly6C$^+$Ly6G$^-$) and PMN (Ly6C$^{int}$Ly6G$^+$). Alternatively, the CD11b$^+$ cells were plotted in an Ly6C versus MHC-II plot to identify patrolling monocytes (Ly6C$^-$MHC-II$^-$). Regarding the RBC subsets, a Ter119 versus FSC-A plot allows identification of RBCs (i.e. Ter-119$^+$ cells). These cells be plot in a CD71 versus Ter-119 plot to identify immature (Ter119$^+$CD71$^+$) and mature (Ter119$^+$CD71$^-$) cells, or in a CD44 versus FSC-A plot to identify nucleated erythroblasts (pro- and basophilic (I), polychromatic (II), orthochromatic (III) erythroblasts) from nucleated reticulocytes (IV) and enucleated erythrocytes (V).
(TIF)

**S4 Fig. Absolute weight loss of *T. congolense* infected mice.** Absolute weights of *T. congolense* infected wild type (WT, black symbol) and TgAlbCre-IL10$^{-/-}$ (red symbol) mice, when considering (subtracting) the increase in hepatosplenomegaly. Data are represented as mean of 3–5 mice per group ± SEM and are representative of 2–3 independent experiments.
(TIF)

**S5 Fig. During the chronic phase of *T. congolense* infection the RBC composition of the blood and spleen is changed.** (A) Total number of RBCs as well as mature and immature RBCs in the blood of *T. congolense* infected (Day 45 p.i.) mice, which were calculated based on the total blood volume (Fig 6A). WT (black symbol), LysM-IL-10$^{-/-}$ (grey symbol) and TgAlbCre-IL10$^{-/-}$ (white symbol) mic. (B) Total number of RBCs as well as mature and immature RBCs in the spleen of *T. congolense* infected (Day 45 p.i.) mice, Dashed line represents cytokine levels in non-infected animals. Data are represented as mean of at least 3–5 mice per group ± SEM and are representative of 2 independent experiments. (*: p≤0.05, ***: p≤0.005).
(TIF)

**S6 Fig. Hepatocyte-IL10 deficiency does not alter the splenic RBC differentiation during *T. congolense* infection.** Percentage of the different erythroid populations (defined as described in S3B Fig) in spleen of WT (black bar) and TgAlbCre-IL10$^{-/-}$ (open bar) mice at 40 days p.i. Results are representative of 2 independent experiments and shown as mean of 3 individual mice ± SEM.
(TIF)

**S7 Fig. Genotyping profile of TgAlbCre-IL10$^{-/-}$ and LysMCre-IL10$^{-/-}$ mice.** Prior to performing experiments mice were genotyped using the conditions described by the supplier

(Jackson mice). Upper left panel: IL10$^{fl/fl}$ genotyping profile, Upper right panel: LysMCre genotyping profile, Lower panel: TgAlbCre genotyping profile.
(TIF)

## Acknowledgments

We would like to thank Ella Omasta, Marie-Therese Detobel, Maria Slazak, Yvon Elkrim and Nadia Abou for technical and administrative assistance as well as Prof. Dr. Geert Raes for the constructive discussions.

## Author Contributions

**Conceptualization:** Benoit Stijlemans, Patrick De Baetselier.

**Data curation:** Benoit Stijlemans, Hannelie Korf, Lea Brys.

**Formal analysis:** Benoit Stijlemans, Hannelie Korf, Lea Brys, Carl De Trez.

**Funding acquisition:** Patrick De Baetselier, Jo A. Van Ginderachter, Stefan Magez.

**Investigation:** Benoit Stijlemans, Hannelie Korf, Carl De Trez.

**Methodology:** Benoit Stijlemans.

**Project administration:** Benoit Stijlemans, Stefan Magez.

**Resources:** Jo A. Van Ginderachter, Stefan Magez.

**Supervision:** Benoit Stijlemans, Patrick De Baetselier, Jo A. Van Ginderachter, Stefan Magez, Carl De Trez.

**Validation:** Benoit Stijlemans, Carl De Trez.

**Visualization:** Benoit Stijlemans.

**Writing – original draft:** Benoit Stijlemans, Patrick De Baetselier.

**Writing – review & editing:** Benoit Stijlemans, Hannelie Korf, Patrick De Baetselier, Jo A. Van Ginderachter, Stefan Magez, Carl De Trez.

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
