## [Decision Letter · Decision Letter 0]

2 Oct 2019

Dear Dr Stijlemans:

Thank you very much for submitting your manuscript "Hepatocyte-derived IL-10 plays a crucial role in attenuating pathogenicity during the chronic phase of T. congolense infection." (PPATHOGENS-D-19-01509) for review by PLOS Pathogens. Your manuscript was fully evaluated at the editorial level and by independent peer reviewers. The reviewers appreciated the attention to an important topic but identified some aspects of the manuscript that should be improved.

We therefore ask you to modify the manuscript according to the review recommendations before we can consider your manuscript for acceptance. Your revisions should address the specific points made by each reviewer.

(1) A letter containing a detailed list of your responses to the review comments and a description of the changes you have made in the manuscript. Please note while forming your response, if your article is accepted, you may have the opportunity to make the peer review history publicly available. The record will include editor decision letters (with reviews) and your responses to reviewer comments. If eligible, we will contact you to opt in or out.

(2) Two versions of the manuscript: one with either highlights or tracked changes denoting where the text has been changed; the other a clean version (uploaded as the manuscript file).

We hope to receive your revised manuscript within 60 days or less. If you anticipate any delay in its return, we ask that you let us know the expected resubmission date by replying to this email.

[LINK]

Sincerely,

Christian R. Engwerda

Section Editor

PLOS Pathogens

David Sacks

Section Editor

PLOS Pathogens

Kasturi Haldar

Editor-in-Chief

PLOS Pathogens

orcid.org/0000-0001-5065-158X

Grant McFadden

Editor-in-Chief

PLOS Pathogens

orcid.org/0000-0002-2556-3526

Reviewer 1 has raised several points that we would like you to address without further experimentation. Please address all their comments and let us know how you have changed the manuscript. Thanks.

Reviewer's Responses to Questions

**Part I - Summary**

Reviewer #1: Here, Stijlemans et al. use IL-10 reporter mice to analyze potential unexpected sources of IL-10 during T. congolense infection of mice. These experiments show that hepatocytes from infected mice express IL-10, both in vitro and in vivo. The authors also show that either parasite or parasite products can stimulate hepatocytes to express IL-10 in vitro. The authors explore the biological relevance of hepatocyte-derived IL-10 using a genetic system (Albumin-Cre.IL-10fl/fl). In these in vivo studies, various cellular, inflammatory, and physiologic parameters are compared among T. congolense-infected TgAlbCre-IL-10-/- mice (but not appropriately matched genetic controls that include either mice with WT alleles of IL-10 or IL-10 floxed mice lacking expression of Cre) and either WT and LysM-Cre.IL-10 fl/fl mice. The rationale for focusing on myeloid-derived IL-10, but not T cell-derived IL-10 is not provided. The experiments show that mice lacking hepatocyte-derived IL-10 expression exhibit elevated parasite burdens and accelerated and exacerbated morbidity and mortality, compared to either WT mice or mice lacking myeloid-derived IL-10.

The paper is generally well written, easy to follow and understandable. The key strength of the manuscript is the unexpected phenotype in T. congolense-infected TgAlbCre-IL-10-/- mice. However, key weaknesses include the substantial volumes of correlative data that appear to mainly derive form a time point when the bulk of animals in the TgAlbCre-IL-10-/- cohort have succumbed to infection. Another weakness relates to studies shown in Figure 12. The studies in Figure 12 both lack comprehensive experimental control groups (e.g. anti-MIF-treated WT mice and irrelevant/non-specific Nb reagents) and stand in direct opposition to the primary parasitemia phenotype reported in Figure 1A.

Other questions and concerns are noted in Parts II and III below.

Reviewer #2: T congolense infection is affecting the livestock making it to a serious problem. Protection is associated with an early and strong Th1 response which has to be tightly regulated to prevent immune pathology by hyper inflammation. An already known and very important factor is IL-10. In IL-10 KO mice are suffering from disease symptoms which are similar to affected cattle.What the authors have convincingly shown is that the important source of IL-10 are not immune cells but hepatocytes.This hepatocyte-derived IL-10 can be triggered by trypanosome lysate or soluble VSG.

The manuscript is in general well written and methods are very good described. Several control were undertaken to really verify the key finding that IL-10 is made by hepatocytes and the importance of this IL-10 was known using tissue specific IL-10 ko mice.The consequences of the lack of IL-10 are clearly described. One important factor that papered to be regulated by IL-10 is MIF. Neutralizing MIF can prevent over inflammation in this model.

In conclusion the manuscript addresses an important disease and provide new insides in immune regulation which might also operate in other diseases.

**Part II – Major Issues: Key Experiments Required for Acceptance**

Reviewer #1: 1) The last sentence of the introduction is an overstatement. The data do support that hepatocyte-derived IL-10 is essential, but the authors have not shown that this is the “only” source that is critical. T cell-specific IL-10 KO may very likely result in the same susceptibility phenotype, and the authors have not explored this. This is also relevant to the authors’ lack of clearly defined rationale for experimentally focusing on myeloid-derived IL-10, but not T cell-derived IL-10. That myeloid cells could produce IL-10 is not much of an argument for ignoring T cell-derived IL-10. Have others already used T cell-specific IL-10 deficient genetic systems to show that T cells are (or are not) a biologically relevant source of IL-10 during T. congolense infection? If those experiments have been published, the reference(s) should be provided. More rationale should be provided.

2) Figure 4: All of the "significant" cellular and inflammatory perturbations occur after parasite control is lost in TgAlbCre-IL-10-/- mice (after day 18, Figure 1A) and after at least half the TgAlbCre-IL-10-/- mice are dead (day 45, Figure 1B). The authors should attempt to reconcile these data. Why do TgAlbCre-IL-10-/- mice lose control of the infection after the second week? Why are the TgAlbCre-IL-10-/- mice dying? These analyses of cellular and inflammatory perturbations, which are quite superficial, precede neither the loss of parasite control nor the death of the mice so it is not at all surprising that very sick mice with very high parasite burdens exhibit these perturbations.

Given the critical role of secreted antibody in controlling this parasite, an obvious question relates to whether humoral immunity derailed in the absence of hepatocyte-derived IL-10. Measuring total cells in the blood does not reveal whether critical circuits/networks such as germinal center reactions, plasma cells, B cell activation, antibody secretion, CD4+ T follicular helper cell numbers and function, or Th1 cells are impaired. Minimally, some discussion is warranted. Relevant to this point, Page 19 lines 13-14: there are no data to support that reduced B cell numbers could explain the elevated parasitemia, given that B cell numbers remain identical through day 28 to 38 p.i., which is well after the TgAlbCre-IL-10-/- mice have lost control of the parasite (which occurs on 18 p.i., day according to Figure 1A).

3) Figure 12: These data represent a very interesting and exciting attempt to understand the inflammatory, parasitological, morbidity and mortality phenotypes in observed in TgAlbCre-IL-10-/- mice, compared to WT mice. As described, one cohort of TgAlbCre-IL-10-/- mice was treated on day 30 p.i. with regular injections of a nanobody targeting host MIF. The data in Figure 12 show that neutralizing MIF abrogates the exacerbated anemia phenotype observed in T congolense-infected TgAlbCre-IL-10-/- mice, but has no effect on modulating parasite burdens. Based on these data the authors argue that MIF signaling mediated the phenotype observed in TgAlbCre-IL-10-/- mice. The authors should specify which phenotype.

This lack of precision aside, there are concerns regarding the parasitemia data shown in Figure 12A. According to these results, there are no differences in parasite burden in untreated WT and untreated TgAlbCre-IL-10-/- mice on either day 35, 40, or 45 p.i. (days 5, 10, 15 post-treatment, respectively, as shown on the graph). However, Figure 1A shows that by either day 30, 35, 40, or 45 p.i., the TgAlbCre-IL-10-/- mice should exhibit burdens that exceed the WT mice by 80% or more (TgAlbCre-IL-10-/- mice have substantially elevated parasite burdens by days 18, 34, 40 and 48 as shown in Figure 1A). Thus, the data in Figure 12 stand in clear opposition to the data shown in Figure 1A and undermine the robustness of the model and data sets.

Reviewer #2: No further experiments required

**Part III – Minor Issues: Editorial and Data Presentation Modifications**

Reviewer #1: 1) Typographical error, Page 7 line 5. Extend should be “extent.”

2) Page 8 line 5: Either data should be shown or a reference should be provided to support the statement that, “levels of IL-10 progressively increase during the course of infection…”

3) The authors argue that IL-10 functions in an autocrine manner to potentiate IL-10 production in hepatocytes. This should be shown by either culturing hepatocytes with recombinant IL-10 or blocking IL-19R signaling in the cultures treated with either lysate, parasites, or VSG.

4) The legend for Figure S1 states that TgAlbCre-IL-10-/- mice did not show any detectable IL-10 protein levels. However, the graph shows substantial proteins levels from leukocyte pools from the TgAlbCre-IL-10-/- mice. Please clarify.

5) In Figure 3, the dashed red line should be defined. Do these lines denote the limit of detection of the assay, or levels measures in naïve mouse sera?

6) Why are the mice infected with T. brucei dying? It appears to have nothing to do with parasitemia (Fig. S2A), which therefore would leave inflammatory immunopathology as the likely cause, which similar to the T. congolense model, would be expectedly counteracted by IL-10. Why might hepatocyte-derived IL-10 not matter after day 25-20 in the T. brucei experimental system?

7) Figure 5D: It is not clear that the IFN-g, TNF, MIF and IL-6 serum data collected on day 45 p.i. represent fundamentally new information given that the relative cytokine levels and relationships were established on day 48 as part of Figure 3.

Reviewer #2: The authors showed that also IL-6 is very differently regulated (indeed more than MIF). Some information should be given why they concentrate on MIF and not IL-6. Indeed new therapeutic strategies to modify IL-6 in humans are already near to clinical application.

PLOS authors have the option to publish the peer review history of their article (what does this mean?). If published, this will include your full peer review and any attached files.

Reviewer #1: No

Reviewer #2: No

---

## [Editor Report · Decision Letter 1]

30 Oct 2019

Dear Dr Stijlemans,

We are pleased to inform that your manuscript, "Hepatocyte-derived IL-10 plays a crucial role in attenuating pathogenicity during the chronic phase of T. congolense infection.", has been editorially accepted for publication at PLOS Pathogens. 

Before your manuscript can be formally accepted and sent to production, you will need to complete our formatting changes, which you will receive by email within a week. Please note that your manuscript will not be scheduled for publication until you have made the required changes.

IMPORTANT NOTES

(1) Please note, once your paper is accepted, an uncorrected proof of your manuscript will be published online ahead of the final version, unless you’ve already opted out via the online submission form. If, for any reason, you do not want an earlier version of your manuscript published online or are unsure if you have already indicated as such, please let the journal staff know immediately at plospathogens@plos.org.

(2) Copyediting and Proofreading: The corresponding author will receive a typeset proof for review, to ensure errors have not been introduced during production. Please review the PDF proof of your manuscript carefully, as this is the last chance to correct any errors. Please note that major changes, or those which affect the scientific understanding of the work, will likely cause delays to the publication date of your manuscript. 

(3) Appropriate Figure Files: Please remove all name and figure # text from your figure files. Please also take this time to check that your figures are of high resolution, which will improve the readbility of your figures and help expedite your manuscript's publication. Please note that figures must have been originally created at 300dpi or higher. Do not manually increase the resolution of your files. For instructions on how to properly obtain high quality images, please review our Figure Guidelines, with examples at: http://journals.plos.org/plospathogens/s/figures.

(4) Striking Image: Please upload a striking still image to accompany your article if one is available (you can include a new image or an existing one from within your manuscript). Should your paper be accepted, this image will be considered for our monthly issue image and may also appear on our website to feature your article. Please upload this as a separate file, selecting "striking image" as the file type upon upload. Please also include a separate "Other" file with a caption, including credits and any potential copyright information. Please do not include the caption in the main article file. If your image is from someone other than yourself, please ensure that the artist has read and agreed to the terms and conditions of the Creative Commons Attribution License at http://journals.plos.org/plospathogens/s/content-license. Please note that PLOS cannot publish copyrighted images.

(5) Press Release or Related Media: If your institution or institutions have a press office, please notify them about your upcoming paper at this point, to enable them to help maximize its impact. If they will be preparing press materials for this manuscript, please inform our press team in advance at plospathogens@plos.org as soon as possible. We ask that you contact us within one week to plan ahead of our fast Production schedule. If you need to know your paper's publication date for related media purposes, you must coordinate with our press team, and your manuscript will remain under a strict press embargo until the publication date and time. This means an early version of your manuscript will not be published ahead of your final version. 

(6)  PLOS requires an ORCID iD for all corresponding authors on papers submitted after December 6th, 2016. Please ensure that you have an ORCID iD and that it is validated in Editorial Manager.  To do this, go to ‘Update my Information’ (in the upper left-hand corner of the main menu), and click on the Fetch/Validate link next to the ORCID field.  This will take you to the ORCID site and allow you to create a new iD or authenticate a pre-existing iD in Editorial Manager

(7) Update your Profile Information: Now that your manuscript has been provisionally accepted, please log into Editorial Manager and update your profile, if needed. Go to https://www.editorialmanager.com/ppathogens, log in, and click on the "Update My Information" link at the top of the page. Please update your user information to ensure an efficient production and billing process. 

(8) LaTeX users only: Our staff will ask you to upload a TEX file in addition to the PDF before the paper can be sent to typesetting, so please carefully review our Latex Guidelines http://journals.plos.org/plospathogens/s/latex in the meantime.

(9) If you have associated protocols in protocols.io, please ensure that you make them public before publication to guarantee immediate access to the methodological details.

Best regards,

Christian R. Engwerda

Section Editor

PLOS Pathogens

David Sacks

Section Editor

PLOS Pathogens

Kasturi Haldar

Editor-in-Chief

PLOS Pathogens

orcid.org/0000-0001-5065-158X

Grant McFadden

Editor-in-Chief

PLOS Pathogens

orcid.org/0000-0002-2556-3526
---

## [Editor Report · Acceptance letter]

27 Jan 2020

Dear Dr Stijlemans,

We are delighted to inform you that your manuscript, "Hepatocyte-derived IL-10 plays a crucial role in attenuating pathogenicity during the chronic phase of *T. congolense* infection," has been formally accepted for publication in PLOS Pathogens.

Best regards,

Kasturi Haldar

Editor-in-Chief

PLOS Pathogens

orcid.org/0000-0001-5065-158X

Michael Malim

Editor-in-Chief

PLOS Pathogens

orcid.org/0000-0002-7699-2064